# Rab1-AMPylation by Legionella DrrA is allosterically activated by Rab1

Jiqing Du [1,2,8], Marie-Kristin von Wrisberg [3,8], Burak Gulen [1,2], Matthias Stahl [1,7], Christian Pett [4], Christian Hedberg[4], Kathrin Lang [3✉], Sabine Schneider [5✉] & Aymelt Itzen [1,2,6✉]

*Legionella pneumophila* infects eukaryotic cells by forming a replicative organelle – the Legionella containing vacuole. During this process, the bacterial protein DrrA/SidM is secreted and manipulates the activity and post-translational modification (PTM) states of the vesicular trafficking regulator Rab1. As a result, Rab1 is modified with an adenosine mono-phosphate (AMP), and this process is referred to as AMPylation. Here, we use a chemical approach to stabilise low-affinity Rab:DrrA complexes in a site-specific manner to gain insight into the molecular basis of the interaction between the Rab protein and the AMPylation domain of DrrA. The crystal structure of the Rab:DrrA complex reveals a previously unknown non-conventional Rab-binding site (NC-RBS). Biochemical characterisation demonstrates allosteric stimulation of the AMPylation activity of DrrA via Rab binding to the NC-RBS. We speculate that allosteric control of DrrA could in principle prevent random and potentially cytotoxic AMPylation in the host, thereby perhaps ensuring efficient infection by Legionella.

[1] Center for Integrated Protein Science Munich (CIPSM), Department of Chemistry, Technical University of Munich, Garching 85748, Germany. [2] Center for Experimental Medicine, Institute of Biochemistry and Signal Transduction, Universitätsklinikum Hamburg-Eppendorf (UKE), Hamburg 20246, Germany. [3] Center for Integrated Protein Science Munich (CIPSM), Department of Chemistry, Technical University of Munich, Garching 85748, Germany. [4] Chemical Biology Center (KBC), Department of Chemistry, Umeå University, Linnaeus väg 10, 90187 Umeå, Sweden. [5] Center for Integrated Protein Science Munich (CIPSM), Department of Chemistry, Ludwig-Maximilians-University Munich, München 81377, Germany. [6] Center for Structural Systems Biology (CSSB), University Medical Centre Hamburg-Eppendorf (UKE), Hamburg, Germany. [7] Present address: Science for Life Laboratory, Department of Oncology-Pathology, Karolinska Institutet, Box 1031, 171 21 Solna, Stockholm, Sweden. [8] These authors contributed equally: Jiqing Du, Marie-Kristin von Wrisberg. ✉email: kathrin.lang@tum.de; sabine.schneider@cup.lmu.de; a.itzen@uke.de

The Gram-negative bacterium *Legionella pneumophila* is the causative agent of Legionnaires' disease. After uptake by human alveolar macrophages via phagocytosis, the pathogen establishes a replicative organelle referred to as the Legionella-containing vacuole (LCV)[1]. The formation and maintenance of the LCV are mediated by ~330 Legionella effector proteins that are released by the bacterial Type IVb secretion system (T4bSS) from the bacterium into the host. These effectors interfere with different host processes, including manipulation of vesicular trafficking (summarised in ref. [2]). In particular, one intensely studied protein, Legionella effector DrrA (Defect in Rab1 recruitment A, also referred to as SidM, substrate of Icm/Dot), is secreted early during infection, and it manipulates the vesicular trafficking regulator Rab1 (Ras-related protein Rab-1).

Rab1 is a member of the Rab family of small GTPases that are involved in spatially and temporally regulating intracellular vesicular trafficking between organelles. Rab proteins act as molecular switches that exist in inactive guanosine diphosphate (GDP) and active guanosine triphosphate (GTP) bound states[3]. At the structural level, the activity state is communicated to interaction partners via two important regulatory loop regions referred to as switch I and switch II. These regions are conformationally flexible in the inactive state but they become structurally ordered in the active form[4]. Only in the active form, Rab proteins mediate signalling via the recruitment of GTP state-specific effector molecules. Guanine nucleotide exchange factors (GEFs) promote GTP loading of Rab, whereas GTPase activating proteins (GAPs) stimulate the intrinsic GTP hydrolysis activity and return Rab back to the inactive state[3]. Furthermore, Rab proteins are post-translationally modified with geranylgeranyl lipids at the structurally flexible C-terminus, enabling reversible membrane attachment[5]. The recycling of geranylgeranylated Rabs from the membrane occurs in the inactive state and is controlled by GDP dissociation inhibitors (GDIs). The biology of Rab proteins has been the topic of a recent review[3].

Due to their pivotal role in regulating vesicular trafficking, Rab proteins are frequently targeted and manipulated by bacterial pathogens. Rab1 controls vesicular transport from the endoplasmic reticulum (ER) to the Golgi apparatus[6,7]. However, it is subjected to intense manipulation during Legionella infections and is rerouted from the ER to the LCV. In this process, Rab1 is known to be targeted by six different Legionella proteins, one of which is DrrA[2,8,9]. However, the deletion of DrrA has little impact on successful infection by Legionella, probably due to effector redundancy[9].

DrrA consists of three functional domains (Fig. 1a). The C-terminal phosphatidylinositol-4-phosphate (PI4P) binding site (referred to as P4M) mediates binding to the PI4P-positive LCV membrane[10,11]. A central Rab1-GEF domain recruits Rab1 via GDP-to-GTP exchange[8,9,12]. The N-terminal domain possesses AMPylation activity (also referred to as adenylylation)[13]. AMPylation is a post-translational modification (PTM) in which an adenosine triphosphate (ATP) is utilised in order to transfer adenosine monophosphate (AMP) to hydroxyl-bearing amino acid side chains. The N-terminal AMP-transferase (ATase) activity of DrrA preferentially mediates AMPylation of active Rab1b at $Y77_{Rab1b}$ (i.e. $Y80_{Rab1a}$ in Rab1a), which is located at the C-terminal end of the important Rab1 switch II region[13,14]. AMPylation inhibits binding to interaction partners such as GDI and molecule interacting with CasL 3 (MICAL3). Also, GAP-catalysed GTP hydrolysis by Rab1 is impaired, thereby rendering the protein permanently bound to GTP when AMPylated[13,15]. Notably, it was reported that GEF-deficient DrrA mutants are competent in the recruitment of Rab1 from Rab1:GDI complex to the LCV[16].

The structure of DrrA has been partially characterised[11–13,17]. However, the structure of the full ATase domain and the

mechanism of Rab1b AMPylation remain elusive. The ATase domain belongs to the DNA polymerase β-like enzyme family and shares structural similarity with the enzyme glutamine synthetase adenylyl transferase (GS-ATase) from *Escherichia coli*[18]. In this class of enzymes, three acidic amino acid residues ($D110_{DrrA}$, $D112_{DrrA}$ and $D150_{DrrA}$) are involved in coordinating an essential $Mg^{2+}$ relevant for ATP binding and AMP transfer. The structural basis for AMP transfer and Rab1 binding has not yet been reported, since structural investigation of Rab1-$DrrA_{ATase}$ complexes is hampered by inherently low affinity; the Michaelis constant ($K_M$) is only ~64 μM[14].

Herein, we employ covalent crosslinking approaches to stabilise and characterise low-affinity $DrrA_{ATase}$:Rab-complexes. The crystal structure of a $DrrA_{ATase}$:Rab-complex reveals an unrecognised regulatory Rab binding site. By mutational analysis, mass spectrometry and activity determinations we show that binding of Rab to this particular position on DrrA acts as a safety switch to control the AMPylation activity of DrrA via an allosteric mechanism in vitro. Our findings lead us to speculate that a potentially harmful AMPylation activity of DrrA may be controlled by Rab1:GTP-binding to an allosteric site.

## Results

**Conceptual design for trapping the DrrA-Rab complex.** Since the ATase domain of DrrA binds to Rab1b only with low affinity[14], we attempted to covalently trap and thereby homogenously enrich and stabilise the complex via two different chemical approaches. We first employed a crosslinking strategy that we recently developed based on site-specific incorporation of unnatural amino acids (UAAs) bearing bromoalkyl moieties (e.g. BrC6K, Fig. 1b) that react specifically with nucleophilic natural amino acids (e.g. Cys, Asp and Glu) in a proximity-enhanced manner upon complex formation[19–21]. The main effect of proximity consists in increasing the effective local concentration of the reactants to boost reaction rates and enable reactions that would not yield products in the absence of the concentration effect. To stabilise the $DrrA_{ATase}$-Rab1b interface, we over-expressed various C-terminally His6-tagged Rab1b constructs (amino acids 3−174; $Rab1b_{3−174}$ with a $Q67A_{Rab1b}$ mutation that minimises Rab1b GTP hydrolysis)[22,23] bearing BrC6K at different positions ($R69_{Rab1b}$, $T72_{Rab1b}$, $I73_{Rab1b}$ and $R79_{Rab1b}$) in the vicinity of the AMPylation site ($Y77_{Rab1b}$), together with an N-terminally Strep-tagged wild type (wt) $DrrA_{ATase}$ construct (amino acids 16−352; $DrrA_{16−352}$). Co-expression of Rab1b with BrC6K at position $69_{Rab1b}$ and wt $DrrA_{ATase}$ resulted in the formation of a covalently crosslinked complex, as confirmed by sodium dodecyl sulphate polyacrylamide gel electrophoresis (SDS-PAGE) shift analysis via α-His6 and α-Strep western blotting (Fig. 1c and Supplementary Fig. 1). We purified the crosslinked complex by affinity and size-exclusion chromatography, and performed tandem mass spectrometry (MS/MS), which indicated that the crosslink corresponds to an ester linkage formed between BrC6K at position $69_{Rab1b}$ within Rab1b and $D82_{DrrA}$ within DrrA (Supplementary Fig. 2 and Supplementary Table 1). This finding was corroborated by mutagenesis experiments, in which the $D82A_{DrrA}$ mutation completely abolished crosslink formation, while mutating the Asp residue with the more nucleophilic Cys led to quantitatively thioether crosslinked Rab1b:DrrA complex (Fig. 1c and Supplementary Fig. 1). Electrospray ionisation MS (ESI-MS) of the purified complex formed in living *E. coli* cells between $R69BrC6K_{Rab1b}$ and $D82C_{DrrA}$ verified the identity of the thioether crosslinked complex, which is AMPylated at $Y77_{Rab1b}$ within Rab1b, confirming the enzyme and substrate activity of both DrrA and Rab1b mutants (Fig. 1d). Intriguingly, amino acid $D82_{DrrA}$ in DrrA is not positioned in the

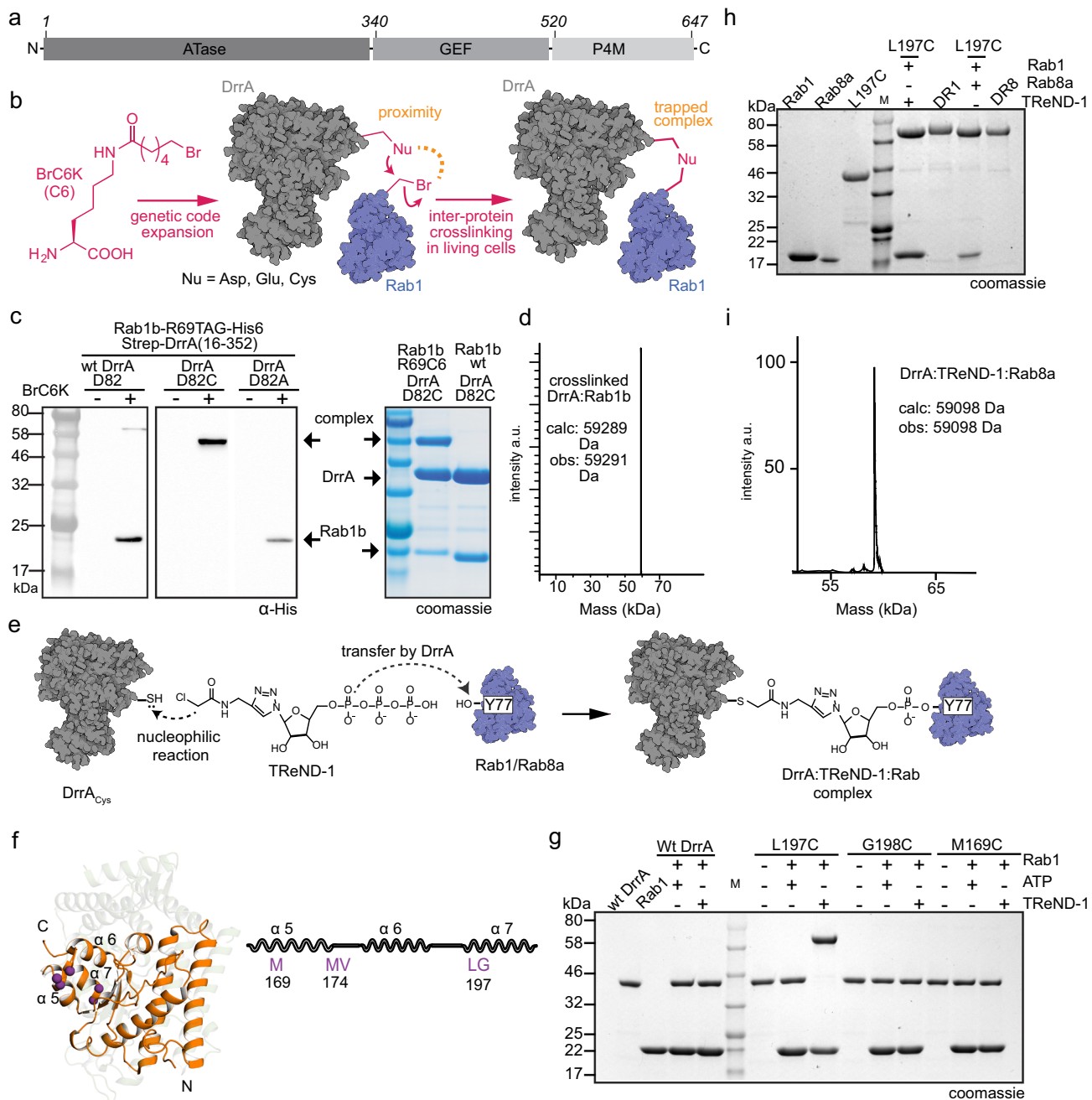

**Fig. 1 Conceptual design for stabilising the low-affinity DrrAATase:Rab complex. a** Schematic representation of domain organisation in full-length DrrA. **b** Covalent stabilisation of DrrA:Rab using site-specifically incorporated UAAs bearing bromoalkyl moieties (BrC6K). **c** α-His6 western-blotting analysis of DrrA:Rab1b complex formation in living *E. coli* cells and SDS-PAGE analysis of in vitro DrrA:Rab1b complex formation. **d** Intact MS analysis of thioether linked AMPylated Rab1b:DrrA$_{D82C}$ complex (a.u.: arbitrary units). **e** Trapping the DrrA:Rab complex via thiol-reactive nucleotide analogues. **f** Selection of Cys substitution sites (M169, M174, V175, L197 and G198, purple) in the model, and superimposition of DrrA$_{8-218}$ (PDB: 3NKU, orange) onto GS-ATase (PDB: 3K7D, light green background). **g** SDS-PAGE shift analysis of DrrA:TReND-1:Rab1b complex formation. **h** SDS-PAGE shift analysis of preparative DrrA:TReND-1:Rab1b (DR1)/Rab8a (DR8) complex formation. **i** Intact mass spectrometry analysis of the DrrA:TReND-1:Rab8a complex. Source data are provided as a Source Data file.

vicinity of the catalytically important amino acids D110$_{DrrA}$ and D112$_{DrrA}$[13] but rather on a surface patch located opposite to the active site of DrrA$_{ATase}$, indicating either complete rearrangement of the previously characterised DrrA fold or an alternative mechanism for enzyme activation. Importantly, full-length D82C$_{DrrA}$ including the GEF and P4M domains (amino acids 1−647; DrrA$_{1-647}$) also displayed efficient crosslinking upon co-expression of R69BrC6K$_{Rab1b}$ (Supplementary Fig. 1). In order to

gain structural insight into Rab1b binding to DrrA$_{ATase}$, we generated several milligrams of thioether-crosslinked DrrA$_{16-352}$:Rab1b$_{3-174}$ complex and subjected them to crystallisation trials.

In parallel, we used a complementary in vitro crosslinking approach to stabilise the low-affinity DrrA$_{ATase}$:Rab interface using our recently developed thiol-reactive nucleotide derivatives (TReNDs) as bridging functionalities[24] (Fig. 1e). TReNDs carry a thiol-reactive chloroacetamide function that can covalently tether

to a strategically placed cysteine in a proximity-enabled manner. In this case, chloroacetamide is positioned at an ATP derivative in which the nucleobase is replaced by a triazole ring. Thus, the combined reaction of the chloroacetamide with a strategically introduced cysteine in DrrA and the AMP transfer by DrrA to Rab proteins enables the formation of a covalently trapped complex between the enzyme and its target protein. Inspired by the identification of $D82C_{DrrA}$ as a site for crosslinking with BrC6K-bearing Rab1b (see above) and by a previous prediction of the ATP-binding site in DrrA[25], we intended to probe both the putative adenine binding pocket and the amino acid residues located at the opposite surface of DrrA for complex formation with TReNDs and Rab1b. Therefore, we generated $DrrA_{16-352}$ constructs bearing M169C, M174C, V175C (the putative adenine interaction interface) and L197C, and G198C (opposite to the putative adenine interaction site) substitutions (Fig. 1f), and tested their ability to form a covalent complex with TReNDs and $Rab1b_{3-174}$ loaded with the non-hydrolysable GTP analogue GppNHp. TReND-1 was indeed transferred by wt $DrrA_{16-352}$ to $Rab1b_{3-174}$, indicating that it is correctly positioned in the catalytic centre (Supplementary Fig. 3). Using recombinantly purified proteins, TReND-1 successfully formed a covalent ternary complex with $Rab1b_{3-174}$ and $DrrA_{16-352}$-L197C, as demonstrated by an apparent increase in molecular weight in denaturing SDS-PAGE (Fig. 1g), while ATP failed to form a covalently linked ternary complex, as expected. The reaction was site-specific since other mutants did not produce a covalent product (Fig. 1g and Supplementary Fig. 4). Importantly, residue $L197_{DrrA}$ is also not located in the vicinity of the ATase active site; it is situated on the same surface patch as residue $D82_{DrrA}$ opposite the catalytically active amino acid residues of DrrA (Supplementary Fig. 1).

Encouraged by the efficiency of our crosslink formation with the $DrrA_{16-352}$-L197C mutant, we attempted to produce larger quantities of the ternary DrrA-TReND-1-Rab1b complex for structural investigations by X-ray crystallography. Several milligrams of pure $DrrA_{16-352}$:TReND-1:$Rab1b_{3-174}$ complex were obtained, but the resulting crystals diffracted only poorly as it was the case for the thioether crosslinked $DrrA_{16-352}$:$Rab1b_{3-174}$ described above. Therefore, we also generated the TReND-1-linked complex with the GTPase domain of the close Rab1b-homologue Rab8a (amino acids 6−176; $Rab8a_{6-176}$; Fig. 1h). Rab1b and Rab8a share 53% sequence identity, their structures can be superimposed with an RMSD (root-mean-square deviation) of about 0.5 Å, and Rab8a can be AMPylated by DrrA in vitro with very similar kinetics (Supplementary Fig. 5)[13]. The integrity, homogeneity, and correctness of the covalently linked ternary DrrA:TReND-1:Rab8a complex were validated using high-resolution (HR) MS of intact proteins, and the results demonstrated excellent accordance between experimental and theoretical mass (Fig. 1i). Thus, combining thiol-reactive UAAs or nucleotides with cysteine-substituted DrrA allowed us to prepare covalent DrrA-Rab complexes, overcoming the difficulties in accessing the transient $DrrA_{ATase}$-Rab complex by more traditional approaches.

**Structure of the DrrA:Rab8a complex**. In order to obtain insight into the molecular basis of DrrA substrate recognition, we determined the crystal structure of the $DrrA_{16-352}$:TReND-1:$Rab8a_{6-176}$ complex bound to GppNHp at 2.1 Å resolution ($DrrA_{16-352}$:TReND-1:$Rab8a_{6-176}$ will be referred to as DrrA:Rab8a hereafter for clarity) (Fig. 2a, Supplementary Table 2 and Supplementary Fig. 6). The complex binding mode observed in the crystal structure was unexpected since the Rab molecule does not interact with the catalytic centre of $DrrA_{16-352}$ constituted by the catalytically important residues $D110_{DrrA}$ and $D112_{DrrA}$[13].

Instead, Rab8a:GppNHp binds to a surface patch opposite the catalytic centre that we refer to as the non-conventional (NC) Rab binding site (RBS), hereafter referred to as NC-RBS (Fig. 2a, d). The complex exhibits the general structural features of the previously determined individual subunits of Rab8a:GppNHp, and parts of $DrrA_{8-218}$ and $DrrA_{210-534}$ (Fig. 2a)[13,17,26]. In accordance with previously solved structures, Rab8a in complex with DrrA has a fold consisting of a central six-stranded β-sheet surrounded by five α-helices[26]. Albeit individual crystal structures of N- and C-terminal parts of DrrA are available[13,17], the structure of the full ATase domain of DrrA has not been reported to date. The general arrangement of the secondary structure elements of the so far solved fragments agrees very well with the full ATase domain, yet significant positional differences in the catalytic site residues were observed. Based on the structures of other ATases belonging to the DNA polymerase β-like enzyme family, the catalytic motif of DrrA is expected to be $G98$-$S99$-$L100$-$X_{11}$-$D110$-$X$-$D112_{DrrA}$, with $D110_{DrrA}$ and $D112_{DrrA}$ involved in coordinating the essential magnesium ion. In contrast to the previous $DrrA_{8-218}$ fragment structure[13], the position of $D110_{DrrA}$ and $D112_{DrrA}$ is well defined and the residues appear to be capable of $Mg^{2+}$ coordination (Fig. 2a).

The covalently bound TReND-1 is well defined in the electron density in the complex crystal structure (Fig. 2b). However, due to structural differences of the chemical linker, bridging the TReND-1 ribose and C197 in DrrA, compared to ATP, only few interactions with the central scaffold of the linker are formed, which is reflected by the higher B-factors in respect to the surrounding amino acid residues (Supplementary Fig. 7). In the DrrA:Rab8a complex structure, switch I ($E30_{Rab8a}$, $F33_{Rab8a}$ and $N34_{Rab8a}$), the interswitch region ($D44_{Rab8a}$ and $K58_{Rab8a}$), and switch II ($Q60_{Rab8a}$ and $R69_{Rab8a}$) of Rab8a mainly interact with the N-terminal half of the ATPase domain of DrrA opposite the catalytic motif, centred at amino acids $R70_{DrrA}$, $Q71_{DrrA}$ and $K74_{DrrA}$. Evaluation using the Protein Interfaces, Surfaces and Assemblies (PISA) web service[27] showed that the solvent-accessible area buried upon complex formation is small (648 Å$^2$), indicating that the complex is of low affinity and requires covalent linkage in order to be observed (Fig. 2c; hydrophobic interactions can be seen in Supplementary Fig. 8). In the previous structure of the N-terminus $DrrA_{ATase}$, the catalytic site ($D110_{DrrA}$, $D112_{DrrA}$ and $D150_{DrrA}$) was disordered and incompetent for $Mg^{2+}$ coordination in this arrangement. However, in our complex crystal structure, all three acidic amino acids are properly positioned upon Rab binding to the NC-RBS, as observed by superimposition with the ATase-domain of glutamine synthetase adenylyl transferase from *E. coli*[18] (Fig. 2e). Therefore, the C-terminal half of $DrrA_{ATase}$ appears to be important for overall structural integrity and correct alignment of the catalytic site.

The structure of free Rab8a:GppNHp superimposes very well with Rab8a in the NC-RBS-complex, with an RMSD of 0.38 Å, indicating that binding of Rab8a to DrrA does not result in global structural changes in Rab8a (Fig. 2f)[26]. Similarly, AMPylated Rab1b also superimposes with Rab8a in the NC-RBS-complex, with an RMSD of about 0.6 Å, demonstrating that binding to the NC-RBS of DrrA is also unlikely to induce significant conformational changes in Rab1b (Fig. 2f)[13].

Consequently, the DrrA:Rab8a crystal structure reveals a previously unrecognised NC-RBS opposite the catalytic centre of the ATase.

**Validation of the non-conventional Rab-DrrA interface**. In order to rule out the possibility that binding of Rab1b or Rab8a to the NC-RBS is an artefact of covalent linkage caused by TReND-1,

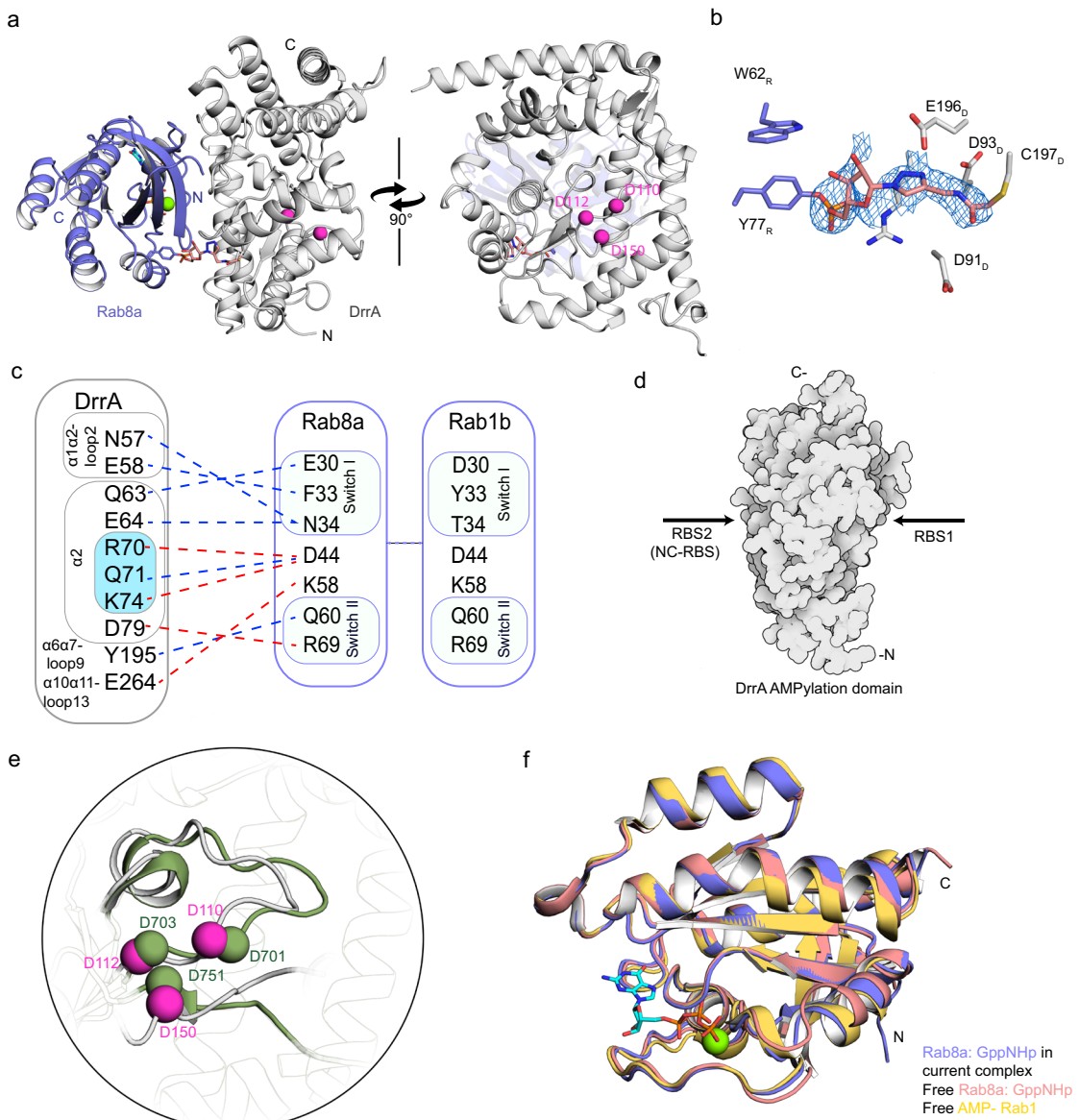

**Fig. 2 Structure of the DrrA:Rab8a complex. a** Orthogonal views of the Rab8-DrrA complex. Pink spheres denote the catalytic Asp residues of DrrA. The green sphere represents the $Mg^{2+}$ ion. **b** Linker density from the unbiased simulated-annealing omit $D$Fo-Fc electron density map contoured at 2.5 σ. The R subscript denotes Rab8a, and the D subscript denotes DrrA. **c** Schematic representation of the Rab8-DrrA interface. Interactions are shown with dashed lines; hydrogen bonds are blue and salt bridges are red. The corresponding interaction residues in Rab1 are shown in the panel on the right. Important residues for maintaining enzymatic activity are coloured cyan, and 'α' represents α-helix. **d** Demonstration of the conventional site (RBS1, containing the catalytic centre) and non-conventional site (RBS2, the back face of the catalytic centre). **e** Structural comparison between the catalytic centre in GS-ATase (PDB: 3K7D, green) and the catalytic centre in DrrA in the DrrA-Rab8a complex. The catalytic centre of DrrA includes $D110_{DrrA}$, $D112_{DrrA}$ and $D150_{DrrA}$. The catalytic centre of GS-AT includes $D701_{GS-AT}$, $D703_{GS-AT}$ and $D753_{GS-AT}$. **f** Structural superposition of free AMP-Rab1:GppNHp (PDB: 3NKV, yellow) and Rab8a (PDB: 4LHW, pink) with Rab8a (blue) from the complex with DrrA. Green spheres indicate $Mg^{2+}$ ions, and GppNHp is shown in stick representation.

we validated the observed complex structure using complementary approaches. First, we produced alanine mutants of DrrA amino acids involved in the interface with Rab8a. Since the Rab-proteins can interact with the NC-RBS and the catalytic site of $DrrA_{ATase}$ at the same time (Fig. 2d), a distinction of the effect of Rab-mutations on AMPylation by binding to either site would not be possible. Hence, we only included DrrA alanine mutations for subsequent experiments. To assess whether the alanine substitutions impaired the general stability of DrrA, we subjected them to thermal unfolding and monitored the change in circular dichroism (CD) signal. Compared with the melting point ($T_m$) of

wt DrrA ($T_m = 60.1$ °C), the other alanine substitutions, but not $D79A_{DrrA}$, displayed mild changes with an amplitude from 0.1 to 2.5 °C, demonstrating that these mutants are biochemically similar to wt $DrrA_{16-352}$ (Fig. 3a). $D79_{DrrA}$ forms intramolecular polar interactions with $K36_{DrrA}$ and $Y40_{DrrA}$ in the complex crystal structure, and $D79A_{DrrA}$ destabilised DrrA, as evidenced by a decrease of 4.2 °C in the melting point.

Since the complex was obtained with Rab8a as a Rab1b-surrogate, we tested whether there are notable differences of the mode of DrrA-binding. Structural superimposition of Rab1b with Rab8a from the DrrA:Rab8a complex revealed four amino acids

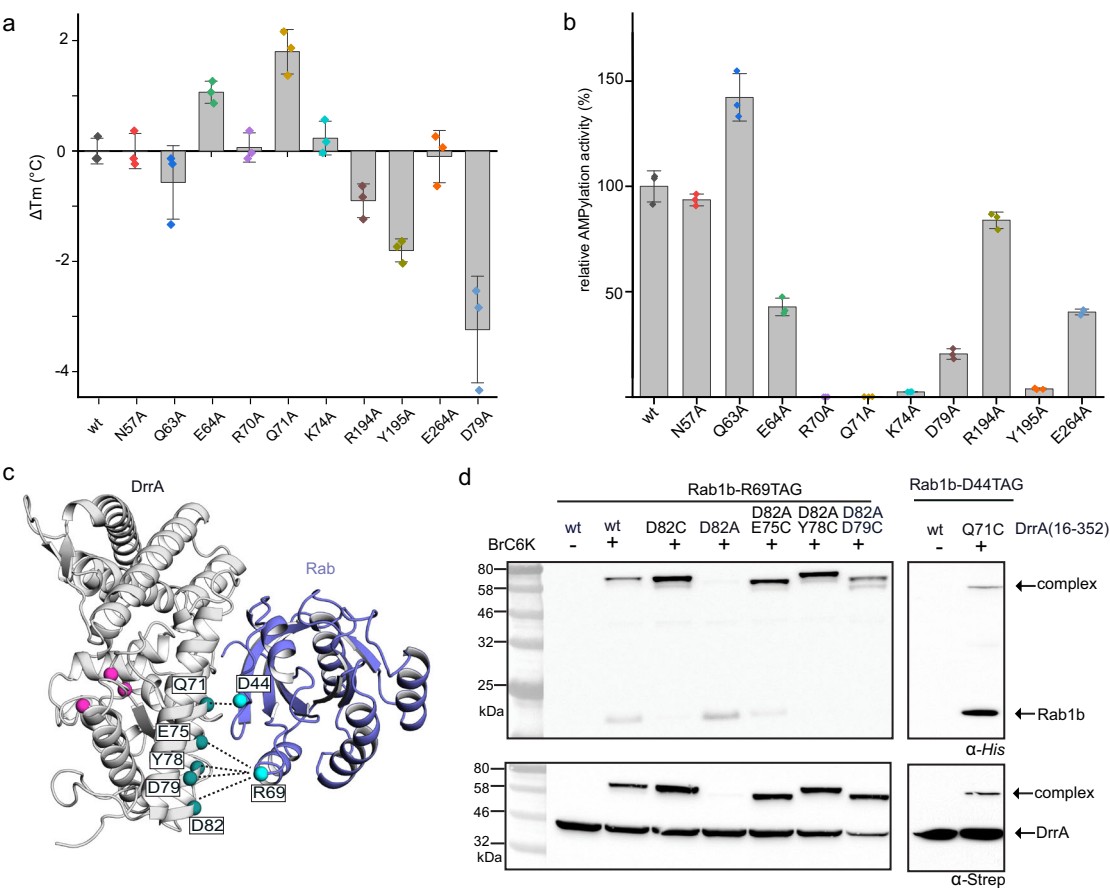

**Fig. 3 Confirmation of the NC-RBS complex interface. a** Melting point (Tm) determination of DrrA and DrrA alanine mutants by circular dichroism (CD). The melting point of wt DrrA is 60.1 °C. Data are means ± standard error of the mean (SEM) from three independent experiments. **b** Determination of Rab1b-AMPylation rates in vitro by DrrA with mutations in the non-conventional site. The $k_{cat}/K_M$ value of wt DrrA is $8.0 \times 10^5$ $M^{-1}$ $s^{-1}$ (± $2.0 \times 10^4$ $M^{-1}$ $s^{-1}$). Data are means ± standard error of the mean (SEM) from three independent experiments. **c** Representation of selected interactions for in cellulo crosslinking in *E. coli*. Pink spheres are the catalytic Asp residues of DrrA. **d** Pairwise BrC6K (within Rab1b) and Cys mutations (within DrrA) based on the DrrA:TReND-1:Rab complex lead to covalent crosslink formation, if the BrC6K-based thioether linker is able to bridge the Rab1b-DrrA interface, as observed by α-His6 and α-Strep western-blotting. Source data are provided as a Source Data file.

that are located in the interface and differ between the Rab-proteins[13]: L26, A29, T32, E35 vs. Rab8a: F26, S29, A32, S35. However, none of these amino acids are involved in interactions in the complex interface and do not create steric clashes that could impair complex formation between DrrA$_{16-352}$ and Rab1b (Fig. 2c and Supplementary Fig. 9). Consistent with the lack of steric clashes and the high sequence homology between Rab1b and Rab8a, the AMPylation rates of these GppNHp-loaded Rabs by DrrA are comparable (Supplementary Fig. 5). Thus, the mode of binding of DrrA$_{16-352}$ to Rab1b:GppNHp and Rab8a:GppNHp is identical.

Next, we investigated the AMPylation activity of these DrrA mutants toward GppNHp-bound Rab1b and Rab8a. AMP-transfer results in the decrease of GTPase tryptophan fluorescence and thus can be used for monitoring AMPylation as reported previously[14]. The DrrA mutants D79A$_{DrrA}$, Y195A$_{DrrA}$ and K74A$_{DrrA}$ exhibited a 5-fold, a 30-fold and a 50-fold decrease in activity toward Rab1, respectively, and R70A$_{DrrA}$ and Q71A$_{DrrA}$ displayed at least 1000-fold activity reduction toward Rab1 (Fig. 3b and Supplementary Fig. 10). The strong decrease in AMP transfer rates can be explained by an ionic and polar interaction network involving R70$_{DrrA}$, Q71$_{DrrA}$ and K74$_{DrrA}$ with D44$_{Rab1b/Rab8a}$. Although Rab8a was used as a surrogate of Rab1b for determining the complex structure, the kinetics of Rab8a-

AMPylation by the DrrA-mutants yields similar results, confirming again an identical mode of DrrA$_{ATase}$-binding for both Rab-proteins (Supplementary Figs. 5 and 10). Therefore, these results suggest that the NC-RBS may play a significant role in regulating DrrA AMPylation activity.

In order to further validate our results, we envisioned testing the complex interface in cellulo using proximity-enabled cross-linking of BrC6K-bearing Rab1b and Cys-mutants of DrrA. Guided by the complex structure, we selected additional combinations for pairwise BrC6K- (within Rab1b) and Cys-substitutions (within DrrA) that we predicted to result in site-specific covalent crosslinking when co-expressing the constructs in *E. coli*. This approach allowed us to screen the interface of the complex within a distance of 13 Å. We confirmed the selective formation of thioether crosslinks between BrC6K incorporated site-specifically at position 69$_{Rab1b}$ within Rab1b and Cys residues introduced at E75$_{DrrA}$, Y78$_{DrrA}$ and D79$_{DrrA}$ within DrrA in living *E. coli* cells using SDS-PAGE, α-His6 and α-Strep western-blotting analyses (Fig. 3c, d and Supplementary Fig. 11). Likewise, crosslinks were observed between D44BrC6K$_{Rab1b}$ and Q71C$_{DrrA}$, as well as between D31BrC6K$_{Rab1b}$ and E280$_{DrrA}$ of wt DrrA. Importantly, the pairwise introduction of BrC6K$_{Rab1b}$ and cysteines$_{DrrA}$ at more remote positions (>15 Å) did not result in covalently linked complexes, indicating that the crosslinking

reaction is specific for proximal sites that can be spanned by the flexible bridge formed between BrC6K and nucleophilic residues (Supplementary Fig. 11).

In summary, our mutational analyses and covalent complex formation studies demonstrate that the interaction of Rab1b with the NC-RBS of DrrA is related to the AMPylation activity of the enzyme, and not an artefact of covalent complex formation using TReND-1 or the UAA BrC6K.

**Rab1-binding to the NC-RBS activates DrrA.** Even though we confirmed the presence of an NC-RBS in the complex crystal structure, the functional relevance of Rab1b binding to the non-catalytic site remains unclear. We first explored whether Rab1b binding causes structural changes in DrrA. Since the apo structure of DrrA has not yet been determined, superimposition of the previously reported N- and C-terminal parts (DrrA$_{8-218}$ and DrrA$_{218-340}$, referred as part-N and part-C respectively hereafter) of the enzyme on our complex crystal structure served as a relevant structural model (Fig. 4a)[13,17]. Even though part-C and part-N have been structurally characterised separately, they do not appear to be distinct functional units. Rather, intimate contacts between part-N and -C are present, even in the loop spanning amino acids, which links these two units. Also, ATase-fragments lacking part-C or portions thereof do not have AMPylation activity and do not cause AMPylation-mediated cellular toxicity. Instead, part-C interacts with helix α3 and the loop α3-β2, which is localised in the catalytic centre of the ATase loop (Supplementary Fig. 12), suggesting that these interactions are necessary for correct folding of this region and positioning of catalytically relevant amino acids. This view is supported by the fact that most interactions (nine out of ten amino acids) between DrrA$_{16-352}$ with the Rab-protein are formed by part-N (Supplementary Fig. 12 and Fig. 2c). The only amino acid interacting with the Rab from part-C is E264$_{DrrA}$ and its mutation to alanine impairs the AMPylation activity moderately (Fig. 3b). Therefore, the lack of enzymatic activity of constructs lacking part-C cannot be explained by the deletion of essential amino acids involved in Rab-binding. Instead, this fact supports the notion that part-C stabilises regions in part-N necessary for catalysis.

Albeit no global structural differences between the composite putative apo structure and the full ATase domain occur (Fig. 4b), significant structural differences can be observed for the secondary structure elements in the vicinity of the DrrA$_{ATase}$ active site (Fig. 4c). Indeed, upon Rab binding, helix α3 may be pulled into the protein core, and D110$_{DrrA}$ (β2), D112$_{DrrA}$ (β2) and D150$_{DrrA}$ (α4−α5 loop) move together so that the active Mg$^{2+}$ coordinating function of these amino acid residues can be fulfilled. Of note, we previously reported that AMPylation-competent DrrA$_{8-533}$ is not able to bind ATP in the absence of Rab1[25]. Therefore, Rab binding to the NC-RBS of DrrA may lead to rearrangement of the enzyme active site. We therefore wondered whether Rab binding to the NC-RBS may regulate DrrA's AMPylation activity by reordering the catalytic residues. In order to investigate the potential influence of Rab1b binding to the ATase domain of DrrA (i.e. DrrA$_{16−352}$), we analysed the rate of Rab1b AMPylation in dependence on the absolute concentration of active Rab1b:GppNHp by measuring the decrease in Rab tryptophan fluorescence upon AMP modification. We observed a lag-phase in the AMPylation rate at low Rab1b:GppNHp concentrations, indicative of a stimulatory contribution of Rab1b binding at higher concentrations (Fig. 4d). Therefore, fitting the data to a simple Michaelis–Menten model (i.e. to a hyperbolic function) was not feasible; hence, a sigmoidal Hill-type function was instead employed. The cooperativity parameter resulting from the Hill fit was greater than 1 ($n = 1.7 \pm 0.16$), indicating a

major contribution of Rab1b:GppNHp-binding to the rate of AMPylation. DrrA$_{16-647}$ containing all three domains shows an identical kinetic profile ($n = 1.6 \pm 0.20$) (Fig. 4e).

Furthermore, we also tested the stimulatory function of Rab1b binding to DrrA in cellulo in order to exclude the possibility of in vitro artefacts. Since DrrA causes AMPylation-induced cellular cytotoxicity in mammalian cells upon overexpression, the activity of individual DrrA mutants can be determined by quantifying the cell viability[13]. To this end, eukaryotic H1299 cells were transfected with the previously characterised alanine substitutions (R70A$_{DrrA}$, Q71A$_{DrrA}$ and K74A$_{DrrA}$) of an N-terminally eGFP (enhanced green fluorescent protein)-tagged DrrA$_{8−533}$ construct. DrrA$_{8−533}$ was employed since the presence of the GEF domain (DrrA$_{340−533}$) is required to produce active, GTP-loaded Rab1b in the cell. Positively transfected cells were collected by fluorescence-activated cell sorting (FACS) and used for cell viability analysis by the absorbance-based MTS (3-(4,5-dimethylthiazol-2-yl)-5-(3-carboxymethoxyphenyl)-2-(4-sulfo-phenyl)-2H-tetrazolium) assay. DrrA$_{8−533}$ but not the AMPylation-deficient D110A/D112A$_{DrrA}$ mutant caused pronounced cytotoxicity[13]. However, H1299 cells transfected with DrrA$_{8-533}$ or with these selected single alanine mutant R70A$_{DrrA}$, Q71A$_{DrrA}$ or K74A$_{DrrA}$ showed similar cytotoxicity (Fig. 4f). Although kinetics indicated that single alanine mutants (R70A$_{DrrA}$, Q71A$_{DrrA}$ or K74A$_{DrrA}$) dramatically decrease the enzyme activity in vitro, results from experiments in cellulo differ from in vitro studies probably due to the longer duration of the in vivo experiments. To make an approximate comparison, we investigated Rab1b-AMPylation by the DrrA single mutants R70A$_{DrrA}$, Q71A$_{DrrA}$ or K74A$_{DrrA}$ via intact high-resolution mass spectrometry at 72 h incubation time. Extended incubation times of the single mutants resulted in nearly quantitative AMPylation of Rab1b (Supplementary Fig. 13). Structural analysis of the DrrA:Rab8a-complex indicated that these three residues can interact with each other to constitute a triangular network, which may stabilise the interaction with D44 from Rab1b/Rab8a. We therefore reasoned that the simultaneous combination of these mutations is required to result in a reduction in cytotoxicity compared to wt DrrA$_{8-533}$. The triple mutant R70A/Q71A/K74A$_{DrrA}$ (corresponding to the NC-RBS) of DrrA$_{8-533}$ was not cytotoxic (Fig. 4g). These data show that mutations in the NC-RBS affect cellular AMPylation and DrrA-mediated cytotoxicity, indicating that the AMPylation activity of DrrA is regulated via Rab1b:GppNHp binding to the non-catalytic site. To further prove that mutations in the NC-RBS site of DrrA occlude binding of Rab under physiological conditions and thereby affect DrrA activity, we set out to perform our UAA-based, proximity-triggered crosslinking approach in living HEK293T cells. We co-expressed BrC6K-bearing Rab1b (Rab1b-R69BrC6K) together with different non-toxic eGFP-DrrA$_{8−533}$ variants in HEK293T cells. Crosslinking was specific for Rab1b-R69BrC6K and the previously identified D82C mutant of eGFP-DrrA$_{8−533}$. In accordance with cytotoxicity data, the triple alanine mutant eGFP-DrrA$_{8−533}$ (R70A/Q71A/K74A) bearing a D82C mutation was deficient in forming a crosslinked complex with Rab1b-R69BrC6K (Fig. 4h and Supplementary Fig. 14).

In addition, we wondered whether the presence of the AMP group in Rab1b at Y77$_{Rab1b}$ would exclude its binding to the NC-RBS. We therefore utilised our established approach and monitored time-resolved crosslink formation (via D82C$_{DrrA}$) of either AMPylated or non-AMPylated Rab1b-R69BrC6K-His6 with the DrrA$_{16-352}$-D82C-D110A-D112A variant by SDS-PAGE and western blotting. The AMPylation-deficient D110A/D112A DrrA-mutant was used to exclude modification of non-AMPylated Rab1b during the experiment. Indeed, both AMPylated and non-AMPylated Rab1 form a crosslinked complex

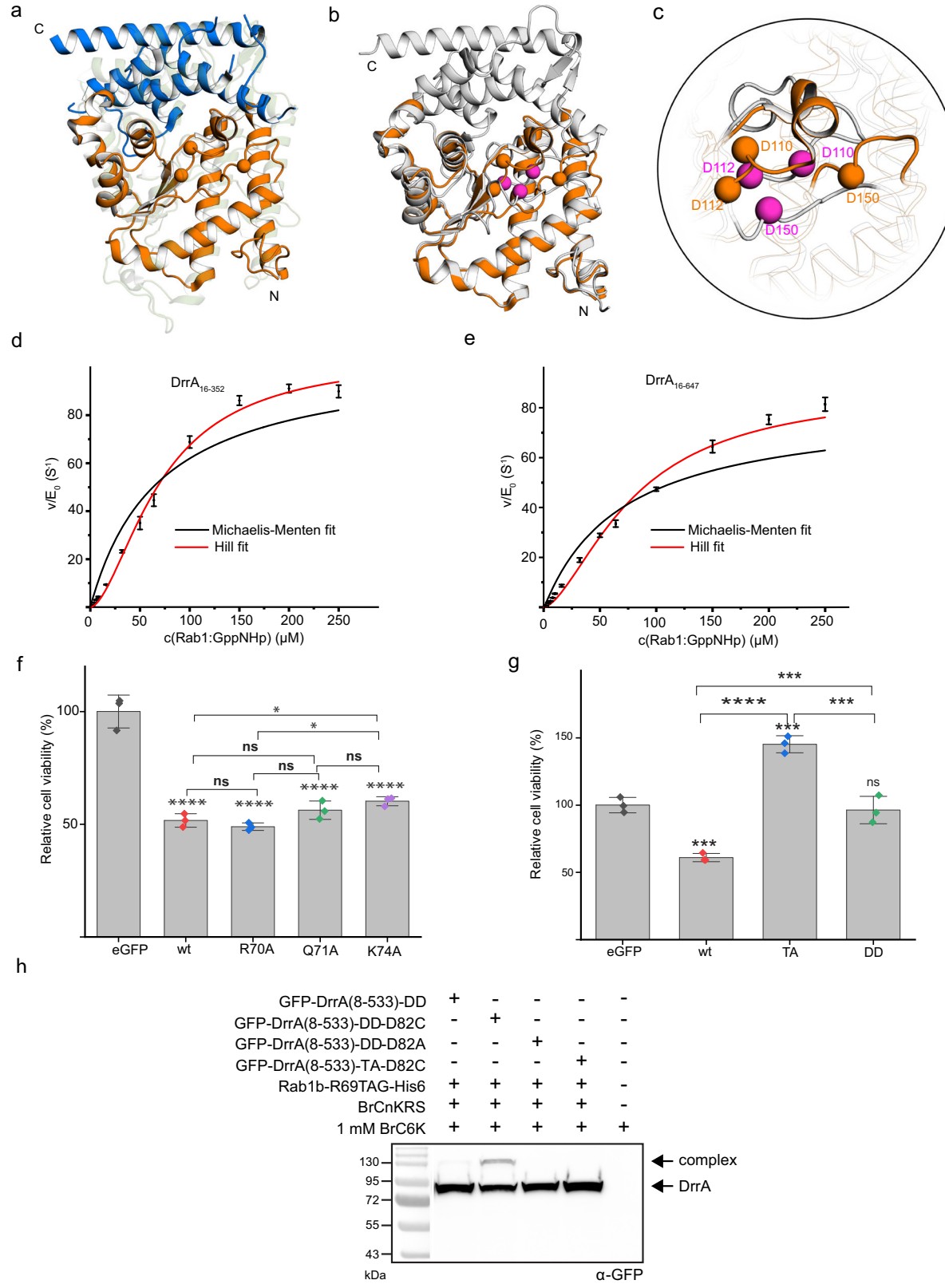

in vitro with minor differences in the rate of reaction (Supplementary Fig. 15). Hence, the presence of AMP in Rab1b does not exclude binding to the NC-RBS.

In summary, biochemical experiments, cytotoxicity data and crosslinking approaches confirm that the AMPylation activity of DrrA is allosterically stimulated by Rab1b-binding to the NC-RBS. However, to what extent the binding of Rab1b to the NC-RBS leads to a structural reorganisation of the catalytic site cannot be deduced entirely from the available structure comparisons: In the absence of the C-terminal part of the ATase-domain (i.e. part-C), essential intramolecular interactions between part-C and part-N are lacking and thus the catalytic

**Fig. 4 Allosteric activation of DrrA by active Rab1. a** Construction of the DrrA apo-form. DrrA$_{8-218}$ (PDB: 3NKU, orange) and DrrA$_{193-352}$ (PDB: 3LOI, blue) are superimposed onto GS-ATase (PDB: 3K7D, light green background). Orange spheres represent the catalytic Asp residues of DrrA$_{8-218}$. **b** Global structural changes in DrrA by binding to active Rab1. DrrA$_{8-218}$ (PDB: 3NKU, orange) and DrrA$_{16-352}$ (grey) are shown. **c** Structural changes in the active centre of DrrA induced by Rab1 binding to the non-conventional site. Purple spheres represent the catalytic centre of DrrA$_{16-352}$ in the DrrA:Rab8a complex. Orange spheres represent the catalytic centre of DrrA$_{8-218}$ (PDB: 3NKU, orange). **d** Sigmoidal dependence of AMPylation on the active Rab1 concentration. The red curve represents the Hill fit with a cooperativity parameter of $n = 1.7 \pm 0.16$; the black curve is the Michaelis–Menten fit. Data are means ± SEM from three independent experiments. **e** Full- length DrrA$_{16-647}$-mediated AMPylation. The red curve represents the Hill fit with a cooperativity parameter of $n = 1.6 \pm 0.20$; the black curve is the Michaelis–Menten fit. Data are means ± SEM from three independent experiments. **f** Cytotoxicity analysis of DrrA single alanine mutants in H1299 cells. Cell viability values (determined by MTS assay) of DrrA-expressing cells (eGFP-positive) were determined in relation to the eGFP vector control. WT: DrrA$_{8-533}$; R70A: DrrA$_{8-533}$_R70A; Q71A: DrrA$_{8-533}$_Q71A; K74A: DrrA$_{8-533}$_K74A. Data are means ± SEM from three independent experiments. One-way analysis of variance (ANOVA) was applied; ns, $p > 0.05$; *$0.01 < p < 0.05$; ****$p < 0.0001$. Comparing to the wt, the $p$ values from R70A, Q71A, K74A are 0.9172, 0.6747, and 0.1594 respectively. The $p$ value between R70A and Q71A is 0.2671, the one between Q71A and K74A is 0.7575, and the one between R70A and K74A is 0.0454. **g** Cytotoxicity analysis of DrrA mutants in H1299 cells. Cell viability values (determined by MTS assay) of DrrA-expressing cells (eGFP-positive) were determined in relation to the eGFP vector control. TA: DrrA$_{8-533}$_R70A_Q71A_K74A; DD: DrrA$_{8-533}$_D110_D112A. Data are means ± SEM from three independent experiments. One-way analysis of variance (ANOVA) was applied; ns, $p > 0.05$; ***$p < 0.001$; ****$p < 0.0001$. Comparing the eGFP, the $p$ values from wt, TA, and DD are 0.0005, 0.0002, and 0.9095, respectively. The $p$ value between wt and DD is 0.0010. The $p$ value between TA and DD is 0.0001. **h** Proximity-triggered crosslinking between BrC6K-bearing Rab1b and eGFP-DrrA$_{8-533}$ mutants in living HEK293T cells as observed by α-GFP WB. Crosslinking is specific for DrrA-D82C and Rab1b-R69BrC6K for DD-DrrA variants; the corresponding triple alanine DrrA-variant (TA-DrrA) is deficient in crosslinking. Source data are provided as a Source Data file.

residues may be more flexible than in a—yet unavailable—ATase reference structure. Thus, it is possible that the impact of the here-observed structural changes introduced by Rab1b on the ATase-domain of DrrA are less pronounced in context of the full-length DrrA protein.

**Formation of a Rab-DrrA complex via the catalytic site**. In addition to assessing the significance of the NC-RBS in DrrA, we attempted to create a complex with the Rab protein bound to the catalytic site of DrrA. For this purpose, we applied the same crosslinking strategy by combining TReND-1 with Cys substitutions located in close proximity to the putative ATP-binding site in DrrA$_{16-352}$. We produced the A176C, D177C and T181C variants of DrrA as potential sites for TReND-1 reaction since these amino acids are likely to be close to the ATP nucleobase interaction site (Fig. 5a). Indeed, DrrA$_{16-352}$ containing the A176C substitution, but not D177C or T181C, formed a covalent ternary complex with Rab1b$_{3-174}$ in vitro, as indicated by an increase in molecular weight observed by SDS-PAGE (Fig. 5b). Using this approach, we were able to obtain milligram amounts of the ternary DrrA$_{16-352}$-A176C:TReND-1:Rab1b$_{3-174}$ complex with high purity, as demonstrated by intact HR-MS (Fig. 5c).

Structure determination of the complex failed due to a lack of diffracting crystals. Nevertheless, the ternary complex from the catalytic site of DrrA permitted us to further validate the presence of NC-RBS of DrrA by evaluating and comparing the catalytic activity. For this purpose, we compared the relative AMPylation activity of DrrA$_{16-352}$ with the ternary complexes linked via the NC-RBS and the catalytic site using the time-resolved change in tryptophan fluorescence of Rab1b in response to AMPylation (Fig. 5d). As expected, the DrrA:Rab1 complex linked via the NC-RBS stimulated the AMPylation activity compared with free DrrA$_{16-352}$. By contrast, the complex linked via the catalytic site was unable to AMPylate Rab1b, showing that the presence of the covalently linked Rab1b blocks the access of substrate molecules. Therefore, our results imply that DrrA$_{16-352}$ contains two separate Rab-binding platforms. Hence, we propose that allosteric binding of active Rab1 to the NC-RBS switches the active site of DrrA$_{ATase}$ from an unstructured AMPylation-deficient state to an organised AMPylation-competent state, which further mediates AMPylation activity (Fig. 5e).

## Discussion

Infection of eukaryotic cells by *L. pneumophila* is a coordinated process characterised by the controlled release and activity regulation of bacterial proteins. The different enzymatic activities of DrrA (GEF and AMPylation) appear to be influenced in particular by the secretion of the protein during the early phases of infection, and the regulated localisation at the LCV[28]. However, in addition to the AMPylation of Rab1 by the N-terminal ATase domain, DrrA can also modify a number of other Rab GTPases, which could cause high cytotoxicity in the case of excessive and mislocalised activity[13]. To avoid these undesirable side effects, which potentially could diminish the chances of an effective infection, additional direct control of the AMPylation activity of DrrA in *L. pneumophila* may be important. Using two different proximity-triggered and site-specific chemical crosslinking approaches employing UAAs bearing alkylbromide moieties or thiol-reactive ATP-derivatives combined with cysteine-substituted DrrA-variants, we discovered a non-conventional Rab1-binding site (NC-RBS) in DrrA. Also, the structure of the DrrA$_{ATase}$ permits us to produce a composite structural model by superimposing previous structures[13,17], in which three different Rab1 binding sites can be visualised (Fig. 5e). The NC-RBS profoundly contributes to the regulation of the AMPylation activity of DrrA in vitro: GTP-loaded Rab1b binds to an allosteric site located opposite to the catalytic centre of the AMPylation domain and stimulates AMP transfer to the GTPase. Thus, two molecules of Rab1b:GTP simultaneously bind to DrrA: one to the allosteric NC-RBS, the other to the catalytic AMPylation site. Hence, DrrA binds to the LCV by virtue of the PI4P binding P4M and catalyses the activation of Rab1 via GDP-to-GTP-exchange using its GEF domain. This displaces Rab1 from GDI and leads to membrane binding through the C-terminally attached geranylgeranyl-lipids. Subsequently, Rab1:GTP could hypothetically stimulate the AMPylation activity of DrrA by binding to the NC-RBS of the ATase-domain, leading to AMPylation of other recruited Rab1 molecules (see model in Fig. 5f). However, this allosteric activation of DrrA has only been shown in vitro and further in vivo work would be necessary to confirm the relevance of this mechanism for infection by *L. pneumophila*.

Interestingly, DrrA is present at the LCV only during the early stages of infection, indicating that it may be released from the membrane to the cytosol as maturation of the compartment progresses[28]. Consequently, there may be a risk of global

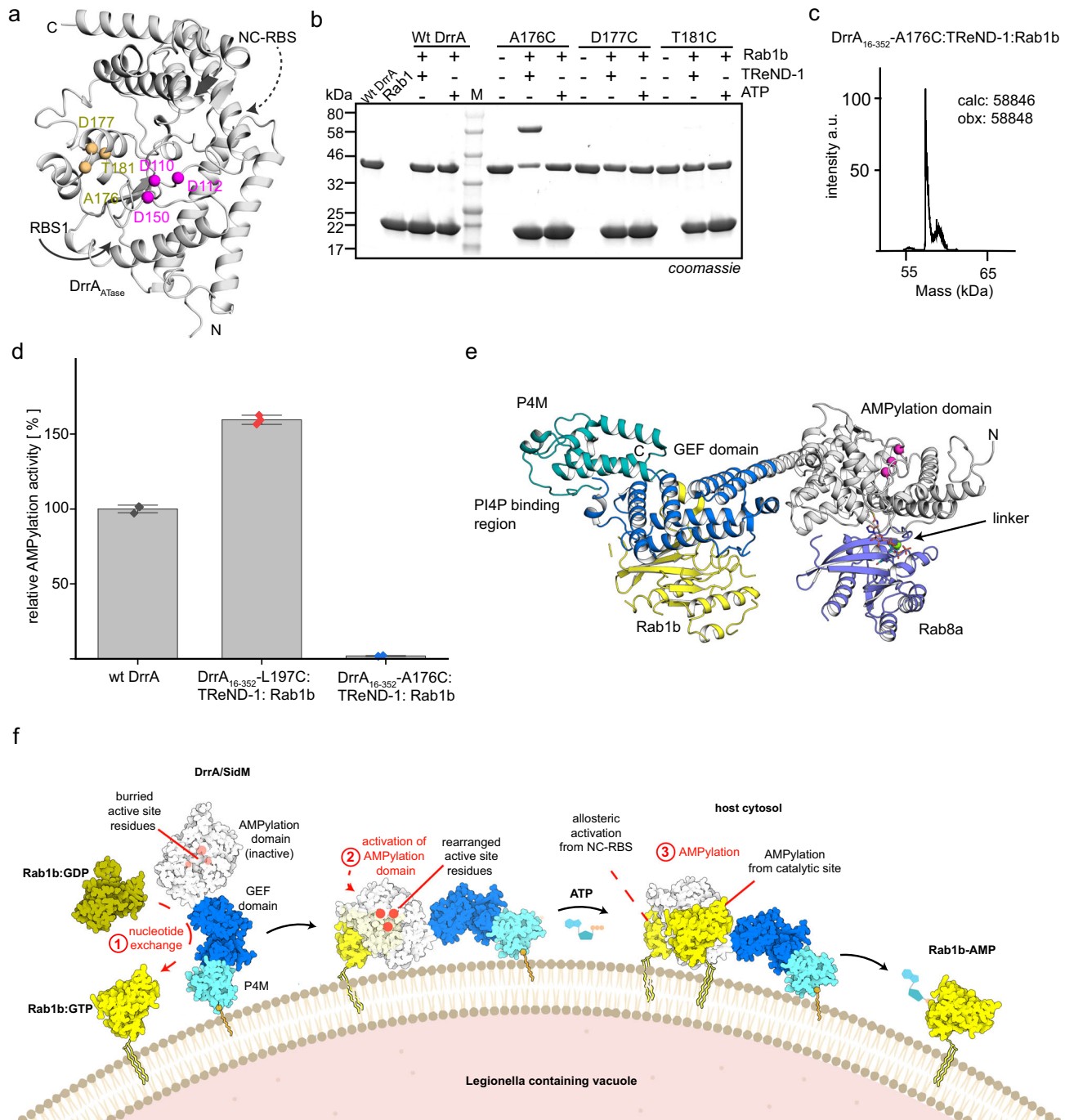

**Fig. 5 Formation of the DrrA:Rab complex in RBS1. a** Cys substitution sites (A176, D177 and T181, light orange) in DrrA$_{ATase}$. Pink spheres represent the catalytic centre of DrrA$_{ATase}$. RBS1 indicates the conventional site, which contains the catalytic centre. NC-RBS represents the non-canonical site, which is located at the back face of the catalytic centre. **b** SDS-PAGE shift analysis of DrrA:TReND-1:Rab1b complex formation in the catalytic side (RBS1). **c** Intact mass spectrometry analysis of the DrrA$_{16-352}$-A176C:TReND-1:Rab1 complex (a.u.: arbitrary units). **d** AMPylation activity comparison of wt DrrA$_{16-352}$ and DrrA:TReND-1:Rab1 complexes using time-resolved tryptophan fluorescence. Data are means ± SEM from three independent experiments. The $k_{cat}/K_M$ value of wt DrrA is $8.0 \times 10^5$ M$^{-1}$ s$^{-1}$ ($\pm 2.0 \times 10^4$ M$^{-1}$ s$^{-1}$). **e** Composite model of DrrA produced from the following crystal structures PDB: 6YX5, 3LOI, 3N6O. The different binding sites of Rab molecules are indicated at the GEF domain and the NC-RBS. The catalytic site of the AMPylation domain (composed by D110, D112, D150) is indicated by magenta spheres. GEF Guanine nucleotide exchange factor; P4M phosphatidylinositol-4-phosphate (PI4P) binding of SidM/DrrA. **f** Model of DrrA action at the membrane. The P4M localises DrrA to the membrane by binding to PI4P. The GEF-domain activates Rab1 by GDP-to-GTP exchange, thereby displacing it from GDI and creating membrane-bound Rab1:GTP (yellow ribbons indicate geranylgeranyl lipids). Binding of Rab1:GTP to the NC-RBS of DrrA putatively results in structural rearrangement of the catalytic site, which stimulates the subsequent AMPylation of another Rab1:GTP-molecule. Source data are provided as a Source Data file.

AMPylation by DrrA with deleterious cellular side effects that may impair infection progression. In this regard, the binding of Rab1b:GTP to a regulatory site in the enzyme perhaps functions as a safety mechanism that limits rogue DrrA activity. Since activated Rab GTPases are exclusively membrane-localised, cytosolic DrrA may not be activated by the cytosolic pool of GDP-bound Rab1b[29]. However, in the GDP form, Rab-proteins are complexed with GDI in the cytosol, thereby making switch II inaccessible for AMPylation by DrrA. But the Rab1:GDI complex is weakly dynamic because cellular GEFs are known to contact switch II, leading to GDI displacement by loading the Rab with GTP. Thus, it is conceivable that DrrA could also AMPylate GDP-bound Rab whenever it has been spontaneously dissociated from GDI, explaining the need for an additional safety mechanism, i.e. allosteric activity control of DrrA-AMPylation activity by Rab1b:GTP-binding. Nevertheless, the hypothesis remains speculative at this point.

There are several instances where Legionella effectors are activated by host proteins. Legionella effector SidJ requires the host protein calmodulin for activation[30,31]. Upon calmodulin binding, SidJ is activated and polyglutamylates SidE (another Legionalla effector) to inhibit the ubiquitin ligase activity of members of the SidE family. This inactivation is required for successful Legionella replication[30,31]. Secondly, Legionella effector VipD lipase requires Rab5:GTP-dependent activation for phospholipase A1 activity-based targeting of enzymes to endosomes, and this eventually inhibits phagosome maturation to ensure the survival of bacteria within the host[32]. Furthermore, there are multiple examples of effector enzymes from other Gram-negative pathogens such as *Salmonella*, *Yersinia* and *Vibrio* that require host protein-dependent activation[33]. After activation by host proteins, these pathogenic effectors cause numerous PTMs such as ADP-ribosylation[34,35], phosphorylation[36,37], acetylation[38] and proteolytic cleavage[39,40]. However, AMPylation that is dependent on activation by a host protein has not been reported previously. Therefore, DrrA comprises a unique example that requires activation by its substrate.

The mode of regulation of the AMPylation activity of DrrA is different from other DNA polymerase β-like enzymes and AMP transferases. For example, AMP-transferring GS-ATase is controlled via binding to the regulatory protein PII, and this molecule binds to a linker region connecting the AMP transferase and AMP removase domains of GS-ATase[18]. Complex formation with PII switches on the AMP transferase site of the enzyme, while turning off the hydrolysis activity of the AMP removase[41,42]. The biochemical and structural modes of regulation are clearly different from those of DrrA as GS-ATase is not dependent on the activity of its substrate glutamine synthetase, and does not contain an allosteric site structurally homologous to DrrA. Other AMP transferases are structurally different from DrrA, and are therefore expected to employ different mechanisms of activity regulation. For example, Fic family enzymes, which are known to AMPylate diverse targets, generally harbour an inhibitory helix that safeguards their activity[43]. However, controlling the positioning or displacement of this inhibitory motif has not yet been demonstrated. Furthermore, in some instances, pseudokinases can possess AMP transferase activity, as exemplified by the highly conserved enzyme SelO[44]. This enzyme is controlled by intramolecular disulfide bond formation, but no regulatory binding partner has yet been identified. Thus, the regulation of DrrA by its substrate is unique among AMP transferases.

In summary, we used complementary approaches for the production of low-affinity DrrA$_{ATase}$:Rab complexes using covalent stabilisation by proximity-enabled reactions. Structural investigation and biochemical characterisation revealed a previously unrecognised regulatory site in DrrA$_{ATase}$ that stimulates AMP transfer via allosteric binding of the protein substrate (i.e. Rab1b:GTP). We speculate that the stimulation of DrrA$_{ATase}$ by Rab1b:GTP confines AMPylation to the site of DrrA localisation and Rab activation (i.e. the LCV), thereby limiting the potential cytotoxicity of the *Legionella* enzyme. It will be interesting to see whether this mechanism can also be demonstrated in vivo during infection by *L. pneumophila*.

## Methods

**Plasmids and reagents.** Full-length *DrrA* was amplified from *L. pneumophila* genomic DNA in previous work[13]. For *E. coli* expression, *DrrA* truncation constructs were cloned into a modified pSF vector (a gift from Stefanie Pöggeler lab, Georg-August-University Göttingen) via Gibson assembly using Gibson Assembly Master Mix (New England Biolabs, Frankfurt, Germany). Constructs for *Rab1* and *Rab8a* were reported previously[13,45]. All point mutations were performed with a Q5 Site-Directed Mutagenesis Kit (New England Biolabs). All plasmids were confirmed by DNA sequencing. Plasmids for genetic code expansion experiments were designed and cloned as described in the Supplementary Information. The synthesis of BrC6K has been reported in detail in Cigler et al.[19]. GppNHp and GDP were purchased from Jena Bioscience, Jena, Germany. The synthesis of TReNDs has been reported previously[24]. Tobacco etch virus (TEV) protease was produced in-house. Lipofectamine LTX for transient transfection in H1299 cells was purchased from Thermo Fisher Scientific (Darmstadt, Germany). MTS agent was purchased from Promega, Walldorf, Germany. Unless otherwise stated, all other reagents were purchased from Sigma (Taufkirchen, Germany) and Carl Roth (Karlsruhe, Germany).

**Molecular biology.** Cloning of constructs for GCE-experiments in bacterial cells: The pBAD-*Rab1b_Q67A-His6* construct was obtained by restriction cloning with HindIII-HF & NdeI (New England Biolabs) and the templates pBad-*His6-TEV-EBFP2* and pMal-*Rab1b 3-174 opti Q67A* and subsequent insertion of the C-terminally His6-Tag via site-directed ligase independent mutagenesis (SLIM) as described by Chiu et al.[46] using the primer pairs P1 & P2 and P3 & P4.

pBAD-RSF1031K-*StrepII-DrrA$_{1-647}$* was cloned with NEBuilder® HiFi DNA Assembly (New England Biolabs). The DrrA$_{1-647}$ gene was amplified using P15 & P16 (template: pET19mod-*DrrA(N451A;R453A;D480A;S483A)*). The backbone fragment was derived by PCR with P13 & P17 (template: pBAD-RSF1031K-*StrepII-DrrA$_{1-339}$*).

pBAD-RSF1031K-*StrepII-TEV-DrrA$_{16-352}$* was achieved by NEBuilder® HiFi DNA Assembly (New England Biolabs). DNA fragments were amplified by PCR using pBAD-RSF1031K-*StrepII-DrrA$_{1-647}$* as template and the primer pairs P18 & P19 and P20 & P21. The assembly yielded the plasmid pBAD-RSF1031K-*DrrA$_{16-352}$*. In a second step the StrepII-Tag and a TEV-site were inserted N-terminally of *DrrA$_{16-352}$* by Q5 site-directed mutagenesis (New England Biolabs) with the primer pair P22 & P23.

The pBAD_Duet-*Rab1b-Q67A-His6_StrepII-DrrA$_{16-352}$* and pBAD_Duet-*Rab1b-Q67A-His6_StrepII-DrrA$_{1-647}$* plasmids were cloned using the NEBuilder® HiFi DNA Assembly (New England Biolabs). The DNA fragments for the assembly reaction were derived by Q5-PCR using the primer pair P24 and P25 for the fragment containing *Rab1b-Q67A-His6* (template: pBAD-*Rab1b-Q67A-His6*) and P26 and P27 for the fragments encoding *StrepII-TEV-DrrA$_{16-352}$* (template: pBAD-RSF1031K-*StrepII-TEV-DrrA$_{16-352}$*) or *StrepII-DrrA$_{1-647}$* (template: pBAD-RSF1031K-*StrepII-DrrA$_{1-647}$*).

Other pBAD-vector or pBAD-Duet-vector variants containing different *Rab1b* and/or *DrrA* mutants were prepared through substitution using Q5 site-directed mutagenesis (New England Biolabs) on the respective pBAD-vector or pBAD-Duet-vector template (Supplementary Table 3).

Cloning of constructs for GCE experiments in HEK293T cells: The p(U6-PylTU25C)4/EF1α-*BrCnKRS* plasmid was derived as described by Cigler et al. in 2017[19]. The p(U6-PylT$_{U25C}$)4/EF1α-*Rab1b-Q67A-R69TAG-His6-4xPylT* was obtained from restriction cloning. The *Rab1b-Q67A-R69TAG_His6* inserts were derived by Q5-PCR with primer M1 & M2 and inserted into p(U6-PylT$_{U25C}$)4/EF1α using NheI-HF and BamHI-HF restriction sites. The pAC-GFP-*DrrA$_{8-533}$-R70A-Q71A-K74A-D82C* plasmid was obtained by Q5 site-directed mutagenesis (New England Biolabs) on pAC-GFP-*DrrA$_{8-533}$-R70A-Q71A-K74A*. The pAC-GFP-*DrrA$_{8-533}$-D110A-D112A-D82A* and pAC-GFP-*DrrA$_{8-533}$-D110A-D112A_D82C* plasmids were obtained by Q5 site-directed mutagenesis (New England Biolabs) on pAC-GFP-*DrrA$_{8-533}$-D110A-D112A*. For pAC-GFP-*DrrA$_{8-533}$-R70A-Q71A-K74A-D82C*, the primer pair M5 & M6 was used. Detailed information for cloning plasmid pAC-GFP-*DrrA$_{8-533}$-D110A-D112A* will be described in the section of Cloning of constructs for MTS-experiments in H1299 cells. For plasmid pAC-GFP-*DrrA$_{8-533}$-D110A-D112A-D82A* and plasmid pAC-GFP-*DrrA$_{8-533}$-D110A-D112A-D82C*, the primer pairs M7 & M8 and M9 & M10 were used, respectively (Supplementary Table 4).

*Cloning of DrrA constructs in DrrA:TReND:Rab complexes study with bacterial cells.* GFP-TEV-DrrA$_{16-352}$ (wt DrrA) was generated by using GFP-StrepII-Tag- TEV-site- DrrA$_{16-352}$ with the primer pair wt-F & wt-R. With this GFP-TEV-DrrA$_{16-352}$

as the template, all the other plasmids are generated by Q5 Site-directed muta-genesis with corresponding primer pairs (New England Biolabs). Rab protein plasmids were cloned before (Supplementary Table 5)[13,47].

*Cloning of DrrA constructs for MTS experiments in H1299 cells.* Plasmid pAC-GFP-DrrA$_{16-352}$ was first prepared with Gibson assembly (New England Biolabs). The primer pair H1 & H2 was used for amplifying the empty pAC plasmid. The DrrA$_{16-352}$ insert was amplified with primer pair H3 & H4. The pAC-GFP-DrrA$_{8-533}$ was produced with the same method. Primer pair H5 & H6 was used for obtaining the DrrA$_{8-533}$ insert fragment.

By using pAC-GFP-DrrA$_{16-352}$ as a template, pAC-GFP-DrrA$_{16-352}$-D110A-D112A was obtained with direct transformation of PCR product (primer pairs H7 & H8). pAC-GFP-DrrA$_{16-352}$-R70A-Q71A-K74A was obtained with the primer pairs H9 & H10 by Q5 Site-directed mutagenesis (New England Biolabs).

For cloning pAC-GFP-DrrA$_{8-533}$-D110A-D112A and pAC-GFP-DrrA$_{8-533}$-R70A-Q71A-K74A, the pAC-GFP-DrrA$_{16-352}$-based plasmids were used for generating corresponding fragments with overhangs to introduce the mutations (primer pairs H5 & H11). Then the empty pAC backbone was amplified for these fragments (primer pairs H12 & H13) (Supplementary Table 6).

**Protein expression and purification.** In general, DrrA constructs are fused to an N-terminal eGFP, which bears an additional 10× histidine tag. In order to further purify DrrA constructs, a TEV cleavage site was introduced for removing the N-terminal eGFP. Also, to facilitate small-scale purification, another Twin Strep tag was inserted between the TEV site and DrrA (His-GFP-TS-TEV-DrrA). Nevertheless, DrrA proteins were expressed in the *E. coli* Rosetta2 (DE3) strain under standard conditions. Briefly, when the OD$_{600}$ of the culture reached 0.6 to 0.8, 1 mM isopropyl β- d-1-thiogalactopyranoside (IPTG) was added to induce protein expression and the temperature was switched from 37 to 20 °C and maintained overnight for expression in Lysogeny broth (LB) medium. DrrA proteins containing Twin Strep tags were purified with Strep-Tactin Magnetic Microbeads (IBA Lifesciences, Goettingen, Germany) by following the manufacturer's protocol. Twin Strep-tag-free DrrA proteins were purified using Ni-NTA resin. TEV protease was then added to cleave the GFP tag, and proteins were further purified by size exclusion chromatography on a 16/600 Superdex 75 pg column pre-equilibrated with 20 mM HEPES, 100 mM NaCl and 1 mM Tris(2-carboxyethyl)phosphine (TCEP).

Rab8a and Rab1b were prepared as described previously[13,48]. Briefly, Rab1b was fused to a N-terminal His$_6$-MBP tag, while Rab8a was fused to a N-terminal His$_6$. Rab1b and Rab8a were expressed in *E. coli* BL21(DE3). When the OD$_{600}$ of the culture reached 0.6 to 0.8, 1 mM IPTG was added to induce protein expression and the temperature was switched from 37 °C to 20 °C and maintained overnight for expression in LB medium. Additionally, Rab8a was co-expressed with the chaperone GroEL/S, and the expression of the GroEL/S chaperone was induced by 1 mg/mL arabinose when OD$_{600}$ was 0.4 to 0.5. After cell disruption using a French press, the lysates were cleared by centrifugation (51,428 g, 40 min, 4 °C), applied to a Ni-NTA resin (buffer A: 50 mM HEPES-NaOH pH 8, 500 mM NaCl, 1 mM TCEP, 1 mM MgCl$_2$, 1 μM GDP) and proteins were eluted with an imidazole gradient (buffer B: buffer A supplemented with 500 mM imidazole). During dialysis with buffer A, the His$_6$-tag was removed by TEV digestion. After reverse Ni-NTA chromatography, Rab proteins were further purified by size exclusion chromatography (buffer: 20 mM HEPES-NaOH pH 8, 100 mM NaCl, 1 mM TCEP, 1 mM MgCl$_2$, 1 μM GDP). Proteins were at least 90% pure. After concentrating to 10 mg/mL, proteins were stored at −80 °C. Nucleotide exchange for Rab1 and Rab8a was performed by following previous protocols. Briefly, 5 mM Ethylenediaminetetraacetic acid (EDTA) was added to Rab1/8a (20 mM HEPES-NaOH pH 8, 100 mM NaCl, 1 mM TCEP, 1 mM MgCl$_2$, 1 μM GDP). A 20-fold excess of GppNHp was then added and incubated at 4 °C overnight. For Rab8a nucleotide exchange assays, an additional 5% (v/v) glycerol was also added for stabilisation, and the buffer was exchanged using NAP columns (GE Healthcare, Munich, Bavaria, Germany). Nucleotide binding to Rab was confirmed by HPLC with a C18 reversed-phase liquid chromatography column.

*Expression and purification of BrC6K-bearing Rab1b-proteins.* Chemically competent *E. coli* NEB 10-beta cells were co-transformed with both pBAD_Rab1b-Q67A-69TAG-His6 (which encodes C-terminally His6-tagged Rab1b gene with an amber codon at position R69) and pEVOL-BrCnKRS-PylT (which encodes *Mm*-BrCnKRS and *Mm*-tRNA$_{CUA}$) plasmids. The transformation was directly inoculated into 50 mL non-AI medium[49] containing ampicillin (100 μg/mL) and chloramphenicol (50 μg/mL) and incubated overnight at 37 °C, 200 rpm. The pre-culture was diluted to an OD$_{600}$ between 0.04–0.07 in 500 mL AI medium[49] supplemented with ampicillin (100 μg/mL), chloramphenicol (50 μg/mL) and nicotinamide (6 mM). The cells were cultivated at 37 °C, 200 rpm to an OD$_{600}$ around 0.2–0.3 before adding BrC6K (2 mM). Cells were expressed for 12 h at 37 °C, 200 rpm.

The cell pellets were resuspended in 30 mL His-wash buffer containing 50 mM HEPES pH 8.0, 500 mM LiCl, 20 mM imidazole, 2 mM β-mercaptoethanol, 1 mM MgCl$_2$ supplemented with 0.1 mg/mL DNase I (AppliChem) and two cOmplete$^{TM}$ Mini EDTA-free protease inhibitor cocktail tablet (Roche). For Rab1b proteins the His-wash buffer further contained 0.01 mM GTP. The cell suspension was incubated on ice for 30 min and sonicated with cooling using an ice-water bath.

The cell lysate was centrifuged (24,000 × g, 1 h, 4 °C) and the cleared lysate was transferred to 1 mL Ni-NTA slurry (Jena Bioscience) and incubated for 1 h at 4 °C slightly shaking. The beads were transferred to a gravity flow column and washed with 200 mL of His-wash buffer before eluting the proteins in 1 mL fractions using His-wash buffer supplemented with 500 mM imidazole pH 8.0. The fractions containing the protein were pooled, concentrated and rebuffered in storage buffer with Amicon® Ultra-4 10 K NMWL centrifugal filter units (Millipore). Rab1b-storage buffer contained 20 mM HEPES pH 8.0, 150 mM NaCl, 5 mM MgCl$_2$, 2 mM DTT and 0.01 mM GTP. DrrA-storage buffer contained 20 mM HEPES pH 8.0, 150 mM NaCl, 1 mM MgCl$_2$, and 2 mM DTT. Purified proteins were analysed by 15 % SDS-PAGE and/or mass spectrometry and stored at −80 °C. Protein concentration was determined using NanoPhotometer® N60 (Implen GmbH).

*Expression and purification of DrrA variants for GCE-based in vitro experiments.* Chemically competent *E. coli* NEB 10-beta cells were transformed with pBAD-RSF1031K-StrepII-TEV-DrrA$_{16-352}$-XXC (which encodes N-terminally StrepII-tagged DrrA$_{16-352}$ gene wildtype or cysteine mutants) and directly inoculated into 50 mL non-AI medium[49] containing kanamycin (50 μg/mL). Next day the pre-culture was diluted to an OD$_{600}$ between 0.04–0.07 in 500 mL AI medium[49] supplemented with kanamycin (50 μg/mL) and incubated for 16 h at 37 °C, 200 rpm.

The obtained cell pellets were thoroughly resuspended in 30 mL of Strep-wash buffer containing 100 mM Tris pH 8.0, 150 mM NaCl, 1 mM EDTA and supplemented with 0.1 mg/mL DNase I (AppliChem) and two cOmplete$^{TM}$ Mini EDTA-free protease inhibitor cocktail tablets (Roche). The cell suspension was incubated on ice for 30 min and sonicated with cooling in an ice-water bath. The lysed cells were centrifuged (24,000 × g, 45 min, 4 °C), the cleared lysate added to 500 μL of Step-Tactin®XT Superflow 50 % Suspension (IBA Lifesciences) and the mixture was incubated for 5 min at 4 °C. After incubation, the Step-Tactin®XT beads were transferred to a gravity flow column and washed with 200 mL of Strep-wash buffer. The proteins were eluted in 1 mL fractions with Strep-wash buffer supplemented with 50 mM biotin. The fractions containing the protein were pooled together, concentrated and rebuffered (20 mM HEPES pH 8.0, 150 mM NaCl, 2 mM DTT, and 5 mM MgCl$_2$) using Amicon® Ultra-4 10 K NMWL centrifugal filter units (Millipore). Purified proteins were analysed by 15 % SDS-PAGE and stored at −80 °C. Protein concentration was determined using NanoPhotometer® N60 (Implen GmbH).

**In vitro crosslinking of Rab1b and DrrA using PEPC via GCE.** Purified proteins of Rab1b variants were mixed with DrrA variants and 1.5-fold excess of Rab1b component (Rab1b:DrrA = 1.5:1) in Xlink buffer (20 mM HEPES pH 8.0, 150 mM NaCl, 2 mM β-ME, 1 mM MgCl$_2$, 0.01 mM GTP) and ATP (final 25 μM) was added. Samples were incubated at 25 °C, 200 rpm and samples were taken at different time points (30 min, 60 min, 120 min, 240 min, O/N). Reactions were stopped by directly adding 4x SDS-Loading buffer and cooking at 95 °C for 3 min. Samples were analysed by SDS-PAGE.

**In vitro crosslinking of Rab1b-R69BrC6K and DrrA-D82C.** Non-AMPylated or AMPylated Rab1b variants (Rab1b3-174-Q67A-R69BrC6K-His6 or Rab1b3-174-Q67A-R69BrC6K-Y77AMP-His6) were mixed with DrrA variants (DrrA16-352-D82C or DrrA16-352-D82C-D110A-D112A) in a 1.5:1 ratio (Rab1b:DrrA = 1.5:1) in Xlink buffer (20 mM HEPES pH 8.0, 150 mM NaCl, 2 mM β-ME, 1 mM MgCl$_2$, 0.01 mM GTP) and ATP (final 25 μM) was added. Samples were incubated at 25 °C, 200 rpm and samples were taken at different time points (30 min, 60 min, 120 min, 240 min, o/N or 0 min, 30 min, 60 min, 150 min, 300 min). Reactions were stopped by directly adding 4x SDS loading buffer and cooking at 95 °C for 3 min. Samples were analysed by SDS-PAGE and α-His6 western blot.

**In cellulo crosslinking of Rab1b:DrrA using BrC6K.** Chemically competent *E. coli* NEB 10-beta cells were co-transformed with either pBAD_Duet-Rab1b-Q67A-XXTAG-His6_StrepII-TEV-DrrA$_{16-352}$-XXC or pBAD_Duet-Rab1b-Q67A-XXTAG-His6_StrepII-DrrA$_{1-647}$-XXC (encoding the C-terminally His6-tagged Rab1b gene with an amber codon at different positions and an N-terminally StrepII-tagged DrrA$_{16-352}$/ DrrA$_{1-647}$ gene with cysteine mutants at different positions) and pEVOL-BrCnKRS-PylT (encoding *Mm*-BrCnKRS and *Mm*-tRNA$_{CUA}$) plasmids[19]. Expression was performed under autoinduction (AI) conditions in 10 mL AI medium[49]. The absorbance of the expression culture at 600 nm (OD$_{600}$) was determined, 1 mL of culture was taken, centrifuged (16,000 × g, 2 min, room temperature), and the supernatant was discarded. According to the OD$_{600}$ value, pellets were resuspended in 1× SDS loading buffer (100 μL of 1× SDS loading buffer was used for 1.0 OD$_{600}$ units). Samples were heated at 95 °C for 10 min, centrifuged (16,000 × g, 15 min, room temperature), and 10 μL was loaded onto a 15% SDS mini gel. After SDS-PAGE, gels were blotted onto a nitrocellulose membrane using an iBlot 2 Dry Blotting system (Invitrogen, Thermo Fisher Scientific) using pro-gramme P0 (20 V for 1 min, 23 V for 4 min, 25 V for 2 min). The membrane was blocked with TBS-T (containing 0.1% (v/v) Tween-20) and 5% (w/v) skimmed milk powder for 1 h at room temperature. The blocking solution was removed and either α-His6-peroxidase antibody (Roche, Penzberg, Germany) or StrepMAB-HRP antibody (IBA Lifesciences, Göttingen, Germany) was added at a 1:5000 dilution with TBS-T containing 1% (w/v) skimmed milk powder. Incubation was

performed overnight at 4 °C, the blot was washed three times with TBS-T, and detection was carried out using a WB Imager Fusion Pulse 6 instrument and Amersham ECL Prime western-blotting Detection Reagent (GE Healthcare, Munich, Germany). Uncropped and unprocessed scans of all gels and blots of this manuscript are provided in the accompanying source data file.

For preparative expression of crosslinked complex, expression was performed as described above in 1 L AI medium[49]. Purification of in vivo crosslinked Rab1b-DrrA complexes involved a two-step procedure comprising Ni$^{2+}$-affinity and size-exclusion chromatography. Cell pellets were resuspended in 30 mL His-wash buffer containing 50 mM 4-(2-hydroxyethyl)-1-piperazineethanesulfonic acid (HEPES) pH 8.0, 500 mM LiCl, 20 mM imidazole, 2 mM β-mercaptoethanol, and 1 mM MgCl$_2$ supplemented with 0.1 mg/mL DNase I (AppliChem, Darmstadt, Germany) and two cOmplete Mini EDTA-free protease inhibitor cocktail tablets (Roche). The cell suspension was incubated on ice for 30 min and sonicated with cooling using an ice-water bath. The cell lysate was centrifuged (24,000 × $g$, 1 h, 4 °C) and the cleared lysate was mixed with 1 mL Ni-NTA slurry (Jena Bioscience) and incubated for 1 h at 4 °C with gentle shaking. The mixture was poured into a plastic column and washed with 200 mL of His-wash buffer before eluting bound proteins in 1 mL fractions using His-wash buffer supplemented with 500 mM imidazole (pH 8.0). Fractions containing the protein were pooled and concentrated with an Amicon Ultra-4 30 K NMWL centrifugal filter unit (Merck KGaA, Darmstadt, Germany) before applying into a Superdex 200 Increase 10/300 GL column (GE Healthcare). Separation was performed at a flow rate of 0.5 mL/min in SEC buffer (20 mM HEPES pH 8.0, 150 mM NaCl, 5 mM MgCl$_2$, 2 mM DTT, 0.01 mM GTP), and 0.5 mL fractions were collected by a 96-well sample collector and analysed using SDS-PAGE (15% mini gels). Fractions containing the complex were pooled, concentrated, and stored at −80 °C or used for MS/MS experiments.

LC-MS of purified full-length proteins was carried out on an Agilent 1260 Infinity Series LC system with an Agilent 6210 ESI Single Quadrupole mass spectrometer using a Phenomenex AerisTM Widepore C4 column (100 × 2.1 mm, 3.6 μm) (Phenomenex, Torrence, USA). Positive mode was used for the analyzation of protein samples and protein UV absorbance at 280 nm was monitored. The protein masses were calculated by deconvolution within the MS OpenLab ChemStation software Edition Rev. C.01.07 SR3 [465] (Agilent Technologies). ProtParam webtool was used for calculation of theoretical protein masses, which were manually corrected with masses for unnatural amino acids.

**MS/MS acquisition of purified crosslinked protein complexes.** Protein samples were lyophilised and resuspended in 200 μL buffer X (7 M urea, 2 M thiourea) for denaturation. To dissolve proteins completely, samples were sonicated for 10 min in an ultrasonic bath. Next, 0.2 μL of 1 M DTT was added to each sample, mixed, and incubated for 45 min at room temperature with shaking at 450 rpm to reduce proteins. For alkylation, 2 μL of 550 mM iodoacetamide (IAA) was added, mixed, and incubated at room temperature with shaking at 450 rpm for 30 min in the dark. The reaction was quenched by addition of 0.8 μL of 1 M DTT and incubation at room temperature for 30 min with shaking at 450 rpm. A 600 μL sample of 50 mM triethylammonium bicarbonate buffer (TEAB) was added so that the pH was ~8. Finally, 1.0 μL trypsin $\left(0.5\frac{\mu g}{\mu L}\right)$ was added and the digest was incubated at 37 °C overnight with shaking at 450 rpm. The reaction was stopped by addition of formic acid (more formic acid was added to ensure the pH remained 3 or below). For desalting, stage-tipping was performed with a double C$_{18}$ membrane using Octadecyl C18 47 mm Extraction disks (Empore Products, CDS Analytical, Oxford, USA). The membrane was washed three times with 70 μL MeOH, 70 μL buffer E (80% (v/v) acetonitrile, 0.5% (v/v) formic acid), and 70 μL 0.5% (v/v) formic acid before the digested protein sample was loaded. The membrane was washed three more times with 70 μL 0.5% (v/v) formic acid before eluting with twice with 30 μL buffer E into a LoBind tube. Samples were lyophilised using a SpeedVac and stored at −80 °C until measurement. Before applying samples to the mass spectrometer, they were dissolved in 30 μL 1% (v/v) formic acid and filtered using Ultrafree-MC-GV centrifugal filters. The filter was washed with 300 μL 1% (v/v) formic acid by centrifugation at 16,000 × $g$ for 2 min before the dissolved sample was filtered (16,000 × $g$; 2 min) and collected into a new LoBind tube.

MS analysis of digested protein samples was performed on an Orbitrap Fusion instrument coupled to an Ultimate 3000 Nano-high-pressure liquid chromatography (HPLC) platform via an electrospray easy source (all Thermo Fisher Scientific). Samples (7 μL) were loaded onto a 2 cm PepMap RSLC C18 trap column (2 μm particles, 100 A, inner diameter 75 μm; Thermo Fisher Scientific) with 0.1% trifluoroacetic acid and separated on a 50 cm PepMap RSLC C18 column (2 μm particles, 100 A, inner diameter 75 μm; Thermo Fisher Scientific) at a constant temperature of 50 °C. The gradient was 5–32% acetonitrile, 0.1% formic acid (7 min 5%, 105 min to 22%, 10 min to 32%, 10 min to 90%, 10 min wash at 90%, 10 min equilibration at 5%) at a flow rate of 300 nL/min. Survey scans ($m/z$ 300–1500) were acquired in the Orbitrap with a resolution of 120,000, and a maximum injection time of 50 ms, with an automatic gain control (AGC) target of 4e5. Most intense ions with charge states of 4–8 and an intensity threshold of 5e3 were selected for fragmentation by high-energy collisional dissociation (HCD) with a collision energy of 30%. Fragment spectra were again recorded in the Orbitrap with a resolution of 30,000, a maximum injection time of 100 ms, and an AGC target of 5e4. The 'inject ions for all available parallelizable time' option was

enabled. Dynamic exclusion was employed with an exclusion duration of 120 s. The overall cycle time was 5 s.

**Crosslinking experiments in living mammalian cells.** Human embryonic kidney 293 T (HEK293T) cells were cultured in Dulbecco's modified Eagle's medium (Gibco™ DMEM, Thermo Fisher Scientific) supplemented with 10% (v/v) FBS (Biochrom) and 1% Pen-Strep solution (10 mg/mL streptomycin, and 10,000 units of penicillin, VWR) at 37 °C in a humidified chamber with 5% CO$_2$. One day prior to transfection, cells were seeded (3.5 Mio. cells per 100 mm dish) on Poly-L-lysine coated dishes. Fresh complete DMEM, supplemented with 1 mM BrC6K was added directly before the transfection of the HEK293T cells using PEI transfection reagent (Sigma-Aldrich) and a 1:3:1 ratio of *BrCnKRS-PylT*-bearing, *Rab1b-PylT*-bearing and *eGFP-DrrA(8-533)*-bearing plasmids with a total DNA-amount of 10 μg per 100 mm dish. Cells were cultivated for 24-30 h, harvested and lysed (in lysis buffer (50 mM Tris-HCl (pH 8), 150 mM NaCl, 20 mM Imidazole; 1x Protease inhibitor solution (VWR)) using freeze-thaw cycles. Cell lysates were cleared by centrifugation (16,000 × $g$, 15 min, 4 °C) and supernatant was analysed by western-blot analysis using the iBlot™ 2 Dry Blotting system (Invitrogen, Thermo Fisher Scientific) and α-His6-Peroxidase antibody (1:5000; Roche) or α-GFP antibody (1:1000; Santa Cruz Biotechnology; in combination with secondary goat-α-mouse-HRP (Invitrogen) in a 1:500 dilution). The detection was carried out at WB Imager Fusion Pulse 6 instrument (Vilber Lourmat) using the Amersham ECL Prime western-blotting Detection Reagent (GE Healthcare).

**Data analysis and crosslink detection.** Raw mass data files were converted to mzML files using MSConvert in ProteoWizard[19,50]. Crosslink searches were then performed with Kojak software version 1.5.5 (http://www.kojak-ms.org)[51] against a database consisting of Rab1b_Q67A_R69K-DrrA$_{16-352}$, common contaminant proteins downloaded from the Andromeda configuration in the MaxQuant software package[52], and all reverse sequences. The MS1 and MS2 resolution was set to 120,000 and 30,000, respectively. Variable modifications included oxidation on methionine (+15.9949) and AMPylation on tyrosine (+329.0525). Static modifications included carbamidomethylation (+57.02146) on cysteine. Three modifications were allowed per peptide. The fragment bin offset was set to 0 and the size was 0.03. A maximum of three missed cleavages were allowed. The minimum peptide mass was set to 300 Da. The precursor mass tolerance was fixed at 10 ppm and the settings for the fragment tolerance were set to fit high-resolution MS2 data (fragment_bin_offset: 0.0 Th, fragment_bin_size: 0.03 Th). The unnatural amino acid was encoded as lysine in our database, hence we searched for crosslinks (+96.0575) between lysines and cysteines, serines, threonines, aspartates, glutamates, and protein N-termini. Additionally, we searched for monolinked species (+175.9837). Annotated spectra were visualised by Kojak Spectrum viewer[53]. Further assessment including hit selection and statistical analysis was performed using custom R scripts (https://github.com/higsch/crosslinkR). Therefore, the Kojak output files *.perc.inter.txt were loaded and each putative crosslink was evaluated whether it contained Rab1b and DrrA, and contained no decoy protein at all. Crosslinks fulfilling these criteria are hits (Supplementary Table 1). The script was further used to assess the confidence of the crosslink identifications by setting the hit scores into the context of the score distributions of all decoy and target hits (Supplementary Fig. 2). Please refer to Supplementary Note 1 on the analysis of mass-spectrometric data to get an in-depth explanation of the underlying statistics and our considerations for our FDR approach here[51,53].

MS proteomics data, Fasta files, and Kojak configuration files have been deposited at the ProteomeXchange Consortium (http://proteomecentral.proteomexchange.org.) via the PRIDE partner repository[54] under dataset identifier PXD019043.

**In vitro AMPylation of Rab1b-Q67A-R69BrC6K-His6.** Purified (by Ni-affinity chromatography) Rab1b$_{3-174}$-Q67A-R69BrC6K-His6 was mixed with DrrA$_{16-352}$ wildtype in a 50:1 Rab1b:DrrA ratio in AMPylation buffer (20 mM HEPES pH 8.0, 150 mM NaCl, 1 mM DTT, 1 mM MgCl$_2$, 0.01 mM GTP) supplemented with an 2.5 excess of ATP compared to the Rab1b component. Samples were incubated at 25 °C, for 3 h. AMPylated Rab1b (Rab1b$_{3-174}$-Q67A-R69BrC6K-Y77AMP-His6) was purified via size exclusion chromatography using a superdex75 10/300 GL column and SEC buffer (20 mM HEPES pH 8.0, 150 mM NaCl, 1 mM DTT, 1 mM MgCl$_2$, 0.01 mM GTP).

**Analytical DrrA$_{Cys}$:TReND-1:Rab complex formation.** To form the DrrA-Rab1 ternary complex, 50 μM cysteine-modified DrrA mutants were incubated with 200 μM TReND-1, 200 μM Rab1:GppNHp (20 mM HEPES-NaOH pH 8, 100 mM NaCl, 1 mM TCEP, 1 mM MgCl$_2$, 1 μM GppNHp) overnight at 25 °C. Similar to the formation of the DrrA-TReND-1-Rab1 ternary complex, an additional 5% (v/v) glycerol was also added to stabilise Rab8a: GppNHp for formation of the DrrA: Rab8a ternary complex. In vitro ternary complex reaction was confirmed by SDS-PAGE and MS. The DrrA:TReND-1:Rab complexes were measured using high-resolution mass spectra recorded on an Agilent 6230 Series TOF MS instrument, equipped with a Dual AJS ESI ion source and coupled to an Agilent 1290 Infinity II LC system. LC was equipped with an Agilent Poroshell C8 column (2.1 mm × 75 mm, particle size 5 μm). The flow rate was set to 600 μL/min, eluent A consisted of milliQ

H2O + 0.1% formic acid, and eluent B of acetonitrile + 0.1% formic acid. Protein samples were injected (1 μL; 0.1–0.5 mg/mL) and eluted with linear gradient of 5–60% B in 3 min. Mass spectra were recorded in Dual AJS ESI mode with a fragmentor voltage of 250 V and mass range of 100–1700 $m/z$, 2 spectra per second. Acquisition software used was Agilent MassHunter 6200 series TOF/6500 series Q-TOF version B.06.01. Protein mass spectra were analysed with Agilent MassHunter Qualitative Analysis software version B.07.00 with deconvolution in Max Entropy mode (mass steps: 1 Dalton; adduct: proton).

**Preparative DrrA$_{Cys}$:TReND-1:Rab formation and purification.** A 200 μM sample of DrrA$_{16−352}$-L197C, 500 μM TReND-1, and 400 μM Rab1 or Rab8a (GppNHp) were incubated overnight at 25 °C or 20 °C. Once the ternary complexes were formed and confirmed by SDS-PAGE, size-exclusion chromatography was performed using a Superdex 26/600 75 pg column (GE Healthcare) to further purify DrrA:TReND-1: Rab complexes. In order to obtain pure complexes for subsequent experiments (e.g. crystallisation screening), a second run of size-exclusion chromatography was conducted. For preparing the DrrA$_{16-352}$-A176C: TReND-1:Rab1 complex, a 200 μM sample of DrrA$_{16−352}$-A176C, 500 μM TReND-1, and 600 μM His$_{10}$-PreScission-Rab1 (GppNHp) were incubated overnight at 25 °C or 20 °C. Following, this complex was purified using Ni-NTA resin. Subsequently, PreScission protease was added to cleave the His tag in Rab1, and then the complexes were further purified by size exclusion chromatography on a 16/600 Superdex 75 pg column (GE Healthcare).

**Cytotoxicity analysis of DrrA mutants.** In order to evaluate the cytotoxicity of different DrrA mutants, lipofectamine LTX was used for transient transfection of H1299 cells with eGFP, wt DrrA, and various other constructs. At 4 h after transfection, the culture medium was replaced with 10% foetal bovine serum (FBS) supplemented Dulbecco's modified Eagle's medium (DMEM). At 24 h after transfection, GFP-positive cells were sorted in the FACS facility of UKE (using a FACS Aria Fusion, see Supplementary Fig. 16 for gating strategy). Next, 24,000 cells were seeded into three wells of a 96-well plate. After 24 h, MTS assays were conducted for cytotoxicity analysis. Cell viability was determined by calculating the ratio of the indicated sample relative to the eGFP vector control. Data were analysed using Origin 2019b (Origin software Inc, Northampton, MA 01060, USA) and Graphpad Prism v 5.0 (GraphPad Software, San Diego, CA 92108, USA).

**Structure determination of the DrrA-Rab8a complex.** The DrrA$_{16−352}$-L197C: TReND-1:Rab8a$_{6-176}$ complex in 20 mM HEPES pH 8, 100 mM NaCl, 1 mM (TCEP), 1 mM MgCl$_2$, and 1 μM GppNHp was concentrated to 14 mg/ml using a centrifugal filter device (Millipore), and centrifuged to remove debris (16,000 × $g$, 15 min, 4 °C) prior to high-throughput crystallisation screening using commercially available screening solutions (NeXtal DWBlock Suites, Qiagen) and robotics (Phoenix, Art Robbins). Complex crystals were obtained by the sitting-drop vapour diffusion method in 0.1 M BICINE and 2.4 M ammonium sulphate pH 9 at 20 °C. Prior to cryo-cooling and storage in liquid nitrogen, crystals were dipped into reservoir solution supplemented with 2 M lithium sulphate for cryoprotection. Diffraction data were collected at the European Synchrotron Radiation Facility (ESRF, beamline MASSIF-1) and were processed with XDS[55]. Crystals belonged to the P321 space group and the resolution cut-off of 2.15 Å was chosen according to the correlation coefficient of random half datasets (CC (1/2)) at ~50%[56–58]. The structure was solved using the coordinates of the previously reported structures of the N-terminal domain of DrrA$_{17−210}$ (PDB: 3NKU) and Rab8 (PDB: 4LHV) by molecular replacement in PHASER[59]. Following simulated annealing with PHENIX[60,61], peaks of density for residues 210−348 of DrrA16$_{16-352}$, which were not present in the search model, were clearly visible. The model was completed by iterative cycles of manual model building in COOT[62] and restraint and TLS (translation/libration/screw) rigid-body refinement in REFMAC5[63]. Structure optimisation was carried out using the PDBredo server[64]. Data processing and structure refinement statistics are included in Supplementary Table 1. All structural figures were prepared with PyMol (Schrödinger). Crystallographic data for the DrrA-Rab8a complex have been deposited in the Protein Data Bank (https://www.ebi.ac.uk/pdbe/) under PDB accession code 6YX5.

**Temperature-scanning CD measurements.** CD signals were recorded on a Chirascan CD Spectrometer (Applied Photphysics, Leatherhead, Surrey, UK) in a 1 mm cuvette with a bandwidth of 0.5 nm and a response of 0.5 s. Temperature scanning CD measurements were taken at 222 nm at a heating rate of 1 K/min. Data were analysed using Origin 2019b and evaluated by the Boltzmann equation to obtain melting temperatures.

**AMPylation kinetics of DrrA.** The kinetics of Rab1b:GppNHp AMPylation by DrrA was monitored via the change in intrinsic Rab1b tryptophan fluorescence using an F-2710 fluorescence spectrophotometer (Hitachi, Schaumburg, IL, USA) (excitation wavelength of 297 nm, emission wavelength of 370 nm, excitation slit width of 2.5 nm, an emission slit width of 5 nm). The Rab1b concentration was <64 μM. However, for Rab1:GppNHp concentration >64 μM, the emission wavelength was shifted to 390 nm, while keeping other parameters constant. All measurements were conducted in the presence of 100 nM DrrA and 1 unit of

pyrophosphate (New England Biolabs) at 25 °C with GppNHp in buffer (20 mM HEPES, 100 mM NaCl, 1 mM MgCl$_2$, 1 μM GppNHp, 1 mM TCEP, pH 7.5) (Supplementary Fig. 17). Data evaluation was performed as previously described[14]. Initial velocities of Rab1b-AMPylation were obtained by fitting the AMP-Rab1b versus time to equation 1:

$$c_{AMP-Rab1}(t) = v * t \qquad (1)$$

where $v$ is the initial velocity, $c_{AMP-Rab1b}(t)$ indicates the concentration of AMP-Rab1b at time $t$ and $t$ is time. For yielding the initial velocity ($k_{obs}$), the initial velocity $v$ was divided by the enzyme concentration.

Hyperbolic curve fitting was performed according to equation 2:

$$k_{obs} = (S * k_{cat}) * (S + K_M)^{-1} \qquad (2)$$

where $S$ is the initial concentration of Rab1b, $k_{cat}$ is the turnover number and $K_M$ is the Michaelis constant.

Sigmoidal curve fitting was performed according to equation 3:

$$k_{obs} = k_{cat} * S^n * (S^n + K_M)^{-1} \qquad (3)$$

where $n$ is the cooperativity parameter.

To determine the AMPylation rates of different DrrA constructs, time-resolved tryptophan fluorescence was applied. In general, 200 μM GTP and 100 nM DrrA$_{GEF}$ were added for the nucleotide exchange of Rab1 (5 μM) from GDP to GTP, and AMPylation was initiated by addition of different DrrA constructs (100 nM) (Supplementary Fig. 17). Data evaluation was performed as previously described[14]. In brief, for the determination of catalytic efficiencies ($k_{cat}/K_M$) of AMPylation reactions measured by fluorescence spectrometry, reaction curves were fitted to a single exponential curve according to using equation 4:

$$F(t) = F_0 + F_A * \exp(-k_{obs} * t) \qquad (4)$$

where $F(t)$ is the fluorescence intensity, $F_0$ is the minimum fluorescence intensity, $F_A$ is the total fluorescence amplitude (i.e., $F_{max}–F_0$, with $F_{max}$ as the maximum fluorescence intensity), and $k_{obs}$ is the observed rate constant. The observed rate constant ($k_{obs}$) was divided by the applied DrrA concentration (100 nM), yielding $k_{cat}/K_M$.

**Reporting Summary.** Further information on research design is available in the Nature Research Reporting Summary linked to this article.

## Data availability
Atomic coordinates and structure factors of the DrrA$_{16-352}$:TReND-1:Rab8a$_{6-176}$ complex have been deposited in the Protein Data Bank with accession code 6YX5. The mass spectrometry proteomics data have been deposited at the ProteomeXchange Consortium (http://proteomecentral.proteomexchange.org/) via the PRIDE partner repository, with data set identifier PXD019043. The data that support the findings of this study are available from the corresponding author upon reasonable request. Source data are provided with this paper.

## Code availability
The R scripts used for hit selection and statistical analysis of the Rab1b_Q67A_R69K-DrrA$_{16-352}$ crosslinking data are available at https://github.com/higsch/crosslinkR.

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

## Acknowledgements

We are grateful to the Knut and Alice Wallenberg Foundation, Sweden (KAW 2013.0187 to C.H.), and the Swedish Research Council (VR) for generous support. We acknowledge support from the collaborative research centre SFB1035 (German Research Foundation DFG, Sonderforschungsbereich 1035, Projektnummer 201302640, projects B05 to A.I. and B10 to K.L.). The authors thank the ESRF and SLS for beam time, and the staff of beamlines MASSIF-1 (ESRF) and PX I (SLS) for assistance with crystal testing and data collection. K.L. is a Professor at TUM-IAS and as such acknowledges funding by the Excellence Initiative and the EU Marie Curie COFUND Program. We would like to thank Johannes Buchner, Matthias Feige and Carolin Rulofs at TU Munich. We are thankful to Stephan Sieber for providing MS infrastructure at TU Munich. We are

thankful for great assistance from other IBS members in UKE, especially Sabine Windhorst, Stefanie Muhs, Gesa König and Vivian Pogenberg. We thank Reinhard Schneppenheim for offering us the H1299 cell line. J.D. was funded by a Chinese Scholarship Council fellowship (2016-2020) for a PhD in TU Munich. B.G. was funded by an Alexander von Humboldt Post-doctoral Fellowship (AvH, 2017-2019). M.S. was funded by the German Academic Scholarship Foundation, as well as the Swedish Childhood Cancer Fund (Barncancerfonden).

## Author contributions

J.D. and B.G. developed the strategy for obtaining the Rab8:DrrA complex. J.D. generated and characterised DrrA mutants, performed kinetic analyses, and analysed DrrA cytotoxicity. M.W. designed and conducted genetic code expansion experiments and in cellulo crosslinking. B.G. assisted writing the manuscript. M.S. identified the UAA crosslink site by mass spectrometry. C.P. measured the TReND-1 based complexes by high-resolution mass spectrometry. S.S. solved and refined the Rab8:DrrA crystal structure. J.D., B.G., S.S., and A.I. analysed and interpreted the complex structure. C.H. designed and contributed TReNDs. K.L., C.H. and A.I. designed the concept of this study. K.L., C.H., S.S. and A.I. provided lab infrastructure. J.D., K.L. and A.I. wrote the manuscript; M.W., M.S., B.G. and C.H. edited the manuscript. A.I., S.S. and K.L. provided oversight over the project.

## Funding

## Competing interests

The authors declare no competing interests.
