## [Peer Review File · Nature Communications]

REVIEWER COMMENTS

Reviewer #1 (Remarks to the Author):

In this study, the authors analyze the regulation of the AMPylation activity of the *Legionella pneumophila* effector DrrA towards human Rab GTPases.

DrrA carries out two biochemical activities towards host Rab1: a GEF activity towards Rab1-GDP, which activates Rab1 by stimulating GDP/GTP exchange, and a post-translational modification activity towards Rab1-GTP (AMPylation). As a result, AMPylated Rab1-GTP is rerouted from the ER to the *Legionella*-containing vacuole (LCV) and is trapped in an active state at the surface of the infectious organelle. The mechanism of Rab1 AMPylation by the ATase domain of DrrA and how DrrA coordinates the GEF and AMPylation activities are currently poorly understood and remain important questions towards understanding how *Legionella* hijacks trafficking host GTPases during infection

Here, the authors designed an elegant strategy to shed new light on certain aspects of these questions. First, using two different targeted crosslinking approaches, they discover a Rab1-binding site located opposite to catalytic center of the ATase domain. The binding site is then depicted in a crystal structure of the Rab1-related Rab8 GTPase bound to the ATase domain of DrrA through crosslinking by a modified nucleotide analog. Finally, the relevance of this unexpected backside interaction is assessed by AMPylation experiments using fluorescence kinetics *in vitro*. Remarkably, the AMPylation efficiency of DrrA is stimulated by increased concentrations of Rab1-GTP, and alanine mutations expected to impair the backside Rab-binding site result in a large decrease in AMPylation rates. This effect is cross-validated by the loss of toxicity in cells of a DrrA construct carrying similar mutations.

Together, this important body of work strongly suggests that Rab1-GTP, the product of the GEF activity of DrrA, functions as an activator of the Rab1 AMPylation activity of DrrA. This depicts a mechanism that has no equivalent in other pathogenic systems targeting small GTPases and constitutes an original advance in the field.

That being said, the structural and biochemical sections of the manuscript do not do full justice to the work and major revisions are needed to fully assess the relevance of the study.

1) The comparison of the ATase domain to previously solved DrrA subdomains should be expanded and better illustrated. Notably, how do the subdomains interact with each other? Do they make contacts in the vicinity of the active center? Is it possible that the Rab1-binding site is comprised of residues from both subdomains? Could that explain why the previously solved truncated ATase domain carrying the active center is inactive and the active center was disordered in this construct?

2) Do both subdomains contact Rab-GTP at the backside site? Does the crosslinker contribute to the interface? Where is the membrane-facing side of Rab1 located in the crosslinked complex?

3) An important question is whether the backside Rab1-binding site can accommodate/be regulated by unmodified Rab1-GTP (which is the product of the GEF reaction), AMPylated Rab1-GTP (which is the product of the AMPylation reaction), or both. An experiment clarifying this issue would provide clues as to the sequence of events in the context of the entire protein and would greatly increase the significance of the study. In addition, the authors should show the docking of unmodified Rab1-GTP and of AMPylated Rab1 within the crosslinked Rab8-DrrA structure to identify possible structural conflicts.

4) Likewise, another important question is whether the proposed regulation of the ATase domain by Rab-GTP also takes place in full length DrrA (which carries the GEF and phospholipid-binding domains in C-terminus of the ATase domain). Kinetics experiments clarifying this issue would increase the significance of the proposed feedforward effect.

5) The biochemical AMPylation experiments should be described more accurately. Notably, controls should be shown that the tryptophan fluorescence signal distinguishes between Rab1 AMPylation and binding of Rab1 to the allosteric site. Also, how are k_{cat}/K_M determined from single rate measurements? In addition, representative kinetics profiles should be shown for Figure 3b (mutants) and Figure 4d (allosteric activation).

6) Cytotoxic effects have been measured in cells with the 70-71-74 DrrA triple mutant. To fully correlate these effects with the *in vitro* AMPylation experiments, the authors should show the toxicity of the same single mutants (57,63,64, 70,71, 74, 194, 195) in cells, which are mentioned but not shown.

7) Could the authors comment on the physiological relevance of the range of Rab concentrations used in Figure 4d (allosteric activation, up to 250 μM)?

8) The proposed regulatory mechanism should take place entirely on the membrane, with the newly discovered interaction of Rab1-GTP with the ATase domain adding an anchoring point. Although assessing this regulatory mechanism in a membrane context would admittedly require too much work, could the authors at least comment on how they envision the structural organization of Rab1 and the three Rab1-binding sites of DrrA on the surface of the LCV, and the sequence of events? A schematic view would be helpful.

9) The proposed structural mechanism for allosteric activation of DrrA is highly speculative and alternative explanations, for example a hinge motion induced by Rab1 to reposition active site residues contributed by the two subdomains, cannot be excluded. This section should be toned down.

10) I am not convinced by the proposed “safety” mechanism, as the switch 2 of cytosolic Rab GTPases is masked by their interactions with RabGDI and is thus unlikely to be able to activate DrrA. Could alternative functions be envisioned?

Minor point: the concentration units in the kinetics methods section contain some errors.

Reviewer #2 (Remarks to the Author):

In this study, Itzen and colleagues continue their biophysical characterization of DrrA (a.k.a. SidM) in its interactions with its substrate Rab1b. The extensive use of biochemical and biophysical developed by the authors, including two different approaches that explore proximity-triggered and site-specific chemical crosslinking for structural analysis of DrrA domains in complex with Rabs has generated some unique insights into the mechanism of catalysis of its AMPylation activity, including the proposal that Rab1b allosterically activates DrrA. Most of the experiments were very well performed, but the evidence for some major conclusions needs further to be substantiated.

Specific comments

1. It is understood that the use of Rab8a for the complex structure is to circumvent the technical challenge in obtaining useful crystals with the bona fide substrate Rab1b. The authors showed that DrrA can AMPylate Rab8a in vitro. Because Rab81 is not the natural substrate it is necessary to

compare the catalysis kinetics of DrrA toward Rab1b and Rab8a. The activity of some mutants showing in Fig. 3 toward these two Rab proteins should also be examined and compared.

2. The physiological relevance of the proposed mechanism of catalysis should be examined. The MPylator activity of DrrA has been shown to be essential for the recruitment of Rab1 to the LCV (PMID: 24520063). It is necessary to introduce some of the mutants in Fig. 3 and Fig. 5b into the *Legionella* drrA mutant and examine their ability in Rab recruitment.

3. The GEF domain of DrrA also contributes to its cytotoxicity to mammalian cells. Results in Fig 5E may have overestimated the effects of the AMPylation activity.

4. The proposed allosteric activation of DrrA by Rab1 is solely supported by conformational changes observed in steady state structures. It is necessary to examine the dynamics of the Rab-induced conformational changes

5. The role of DrrA in *Legionella* virulence was overstated throughout the text. Deletion of drrA did NOT cause discernable defects in bacterial virulence in all infection model tested (Refs 11 and 12). In fact, the drrA gene is absent in *L. pneumophila* strains such as Paris. Some of the references are not accurate. For example, lines 446-447, neither ref 29 nor 30 shows results to demonstrate the importance of SidJ in *L. pneumophila* infection. The study by Gan et al (PMID: 31330531) and an earlier study by Liu and Luo showed that (PMID: 17101649).

6. Line 41 “Legionnaires”.

7. Somewhere in page 4 or 5, the author should discuss the role of the AMPylation activity in Rab1 recruitment to the LCV as shown by Hardiman and Roy (PMID: 24520063).

8. Line 446, “SidE” should be “members of the SidE family” and the authors should reference the study that originally demonstrated the ubiquitin ligase activity of these proteins.

Reviewer #3 (Remarks to the Author):

The manuscript “Rab1-AMPylation by Legionella DrrA is allosterically activated by Rab1” by Du, et al., describes the interaction of Legionella pneumophila DrrA with the host (human) Rab1. Traditional structural identification of the complex failed. However, through amber suppression mutagenesis with crosslinking amino acids and analysis of the resulting complex through MS/MS, x-ray crystallography, and mutagenesis studies, the authors conclude that a non-conventional Rab binding site is used for DrrA binding, resulting in allosteric control of AMPylation as a way to suppress host innate immunity. The crystal structure is solved to sufficient resolution and Rfree levels to correctly identify the interaction interface, although resolution of the linker (Figure 2B) appears to be poor. Mutagenesis studies further corroborate the findings, although the proposed mechanism remains speculative. Two additions are suggested:

1. Supplementary Figure 2 is insufficient. It is large and has low resolution, and the data are not easy to interpret by a non-expert. More careful discussion and analysis of the location of crosslinking is critical to this study and should be provided.
2. B-factors of components of the crosslinked complex (not average) should be analyzed and provided for the proteins and the linker to help clarify the linker resolution

Response to referees

Rab1-AMPylation by Legionella DrrA is allosterically activated by Rab1

We would like to thank the referees for their time and effort. We appreciate their constructive comments on our manuscript. Based on the referee comments we have optimized several aspects of our findings and have particularly improved the presentation of the data and the clarity of our arguments. We hope that the referees will concur with the changes to the manuscript.

We have addressed the reviewer comments below. In order to facilitate reviewing our response, we have applied the following color coding:

Reviewer comments: Black

Author responses: Blue

Text changes in the revised manuscript: Red

Reviewer #1

In this study, the authors analyze the regulation of the AMPylation activity of the *Legionella pneumophila* effector DrrA towards human Rab GTPases.

DrrA carries out two biochemical activities towards host Rab1: a GEF activity towards Rab1-GDP, which activates Rab1 by stimulating GDP/GTP exchange, and a post-translational modification activity towards Rab1-GTP (AMPylation). As a result, AMPylated Rab1-GTP is rerouted from the ER to the *Legionella*-containing vacuole (LCV) and is trapped in an active state at the surface of the infectious organelle. The mechanism of Rab1 AMPylation by the ATase domain of DrrA and how DrrA coordinates the GEF and AMPylation activities are currently poorly understood and remain important questions towards understanding how *Legionella* hijacks trafficking host GTPases during infection

Here, the authors designed an elegant strategy to shed new light on certain aspects of these questions. First, using two different targeted crosslinking approaches, they discover a Rab1-binding site located opposite to catalytic center of the ATase domain. The binding site is then depicted in a crystal structure of the Rab1-related Rab8 GTPase bound to the ATase domain of DrrA through crosslinking by a modified nucleotide analog. Finally, the relevance of this unexpected backside interaction is assessed by AMPylation experiments using fluorescence kinetics *in vitro*. Remarkably, the AMPylation efficiency of DrrA is stimulated by increased concentrations of Rab1-GTP, and alanine mutations expected to impair the backside Rab-binding site result in a large decrease in AMPylation rates. This effect is cross-validated by the loss of toxicity in cells of a DrrA construct carrying similar mutations.

Together, this important body of work strongly suggests that Rab1-GTP, the product of the GEF activity of DrrA, functions as an activator of the Rab1 AMPylation activity of DrrA. This depicts a mechanism that has no equivalent in other pathogenic systems targeting small GTPases and constitutes an original advance in the field.

We appreciate the reviewer's critical evaluation of our manuscript and also thank for the positive reception of our work.

That being said, the structural and biochemical sections of the manuscript do not do full justice to the work and major revisions are needed to fully assess the relevance of the study.

Comment 1

The comparison of the ATase domain to previously solved DrrA subdomains should be expanded and better illustrated. Notably, how do the subdomains interact with each other? Do they make contacts in the vicinity of the active center? Is it possible that the Rab1-binding site is comprised of residues from both subdomains? Could that explain why the previously solved truncated ATase domain carrying the active center is inactive and the active center was disordered in this construct?

As suggested by the reviewer, we have created an illustration describing the role of individually crystallized parts of the DrrA ATase domain (Of note: We prefer not to refer to these parts as subdomains since there is no data that these parts constitute functional domains. Rather, the perception of them being separate results from different constructs used for crystallization). We refer to amino acids 8-218 as the N-terminal part (part-N, color pink) and amino acids 218-340 as the C-terminal part (part-C, color cyan) of the ATase domain (Supplementary Figure 12).

Supplementary Figure 12. Intra- and intermolecular interactions of N- and C-terminal parts of the DrrA_{ATase} domain. a) Schematic representation of N- terminal part (part-N, DrrA₈₋₂₁₈, color pink) and C-terminal part (part-C, DrrA₂₁₈₋₃₄₀, color cyan) in DrrA₁₆₋₃₅₂. b) Specific polar interaction involving sidechain-sidechain or sidechain-backbone between part-N (pink) and part-C (cyan). Interface between the parts is shown in transparent background. c) Interaction map of part-N and part-C of DrrA_{ATase} domain (Laskowski et al., 2018). The residues within the catalytic site are marked with asterisks. Van der Waals contacts are gray, polar contacts are blue, salt bridges are red. d) Mainly the residues in the part-N contacts Rab1 in the back site and only residue which contacts Rab1 in the back site from part-C is E264. Interacting residues are highlighted with black sphere.

The interactions in the interface between the C-terminal part and N-terminal part are mapped as seen by the Supplementary Figure 12a, b, and c. We envision the following model regarding DrrA ATase activity:

1) Part-N and part-C are integral elements of the full ATase domain of DrrA. Their perception as different (sub)domains is derived solely from the artificial and accidental distinction into separate structural units that result from crystallization experiments. Despite of many attempts, the full ATase domain (aa 1-352 and truncation constructs thereof) could never be successfully crystallized, explaining why fragments of this domain have been used for structural characterization instead. However, separately they do not carry any function, even though part-N carries the catalytic center of the ATase domain. Part-N does not show AMPylation activity and does not confer AMPylation dependent cellular toxicity when expressed in mammalian cells (Muller et al., 2010).

Instead, the conserved catalytic motif and neighboring amino acids (DrrA 97-112) require interactions with part-C to properly fold this region into a catalytically competent structure (see Supplementary Figure 12). Therefore, part-C supports the correct arrangement of the catalytic center and thus is necessary for the AMPylation activity.

2) Rab1 likely leads to a further ordering of the catalytic site by binding to the allosteric site. We envision a model where the catalytic residues are generally in place in the absence of Rab1, but binding of the G-protein to the allosteric site leads to a stabilization of the optimal positioning of the catalytic site residues, thereby equipping DrrA_{ATase} with full activity. These hypotheses are supported by the MTS experiments in which allosteric Rab1 binding deficient mutant of DrrA_{ATase} did not cause any cytotoxicity (please see Figure 4g). We have updated the manuscript with the following text to address the comment of the referee:

Even though part-C and part-N have been structurally characterized separately (Muller et al., 2010; Zhu et al., 2010), they do not appear to be distinct functional units. Rather, intimate contacts between part-N and -C are present, even in the loop spanning amino acids, which links these two units. Also, ATase-fragments lacking part-C or portions thereof do not have AMPylation activity and do not cause AMPylation-mediated cellular toxicity (Muller et al., 2010). Instead, part-C interacts with helix $\alpha 3$ and the loop $\alpha 3$ - $\beta 2$, which is localized in the catalytic center of the ATase loop (Supplementary Figure 12), suggesting that these interactions are necessary for correct folding of this region and positioning of catalytically relevant amino acids.

The referee also asked whether the Rab binding site is comprised of residues from part-N and part-C. We have therefore produced an additional supplementary figure that - together with Figure 2c - demonstrates the amino acids of DrrA involved in binding to the Rab-protein (Supplementary Figure 12c and Figure 2c). Out of ten interacting residues in DrrA, only one residue is in the part-C (i.e. E264_{DrrA}) while nine residues are in the N-terminal part. Since E264_{DrrA} forms a salt bridge with K58_{Rab8}, its mutation to alanine affects AMPylation of Rab-proteins by DrrA, though only moderately in comparison to other amino acids mutations located in the allosteric site (see Figure 3b).

This also addresses another comment raised by referee, whether the lack of AMPylation activity after truncation of part-C can be explained by the deletion of amino acids involved in Rab-recognition: Since only one Rab-interacting amino acid (i.e. E264_{DrrA}) is absent in the part-N construct and since the DrrA mutant DrrA_{E264A} still has AMPylation activity, part-C must have more contributions to the enzymatic activity except providing E264_{DrrA} to the Rab binding site. As outlined above, the role of part-C likely is keeping the ATase domain structurally intact and connecting necessary structural elements for catalysis. We have added the following explanation to the manuscript:

This view is supported by the fact that most interactions (9 out of 10 amino acids) between DrrA₁₆₋₃₅₂ with the Rab-protein are formed by part-N (Supplementary Figure 12c and Figure 2c). The only amino acid interacting with the Rab from part-C is E264_{DrrA} and its mutation to alanine impairs the AMPylation activity moderately (Figure 3b). Therefore, the lack of enzymatic activity of constructs lacking part-C cannot be explained by the deletion of essential amino acids involved in Rab-binding. Instead, this fact supports the notion that part-C stabilizes regions in part-N necessary for catalysis.

Comment 2

Do both subdomains contact Rab-GTP at the backside site? Does the crosslinker contribute to the interface? Where is the membrane-facing side of Rab1 located in the crosslinked complex?

Regarding the first comment by the referee: As discussed in the comment 1, Rab:GTP mainly contacts the part-N at the backside site of DrrA. Also, E264 from the part-C contacts K58 of Rab:GTP in the

backside site (Figure 2c). However, this particular interaction proved to be of minor significance, since the E264A_{DrrA} mutant decreases the enzymatic activity only moderately in comparison to other backside amino acid mutations.

Regarding the crosslinker contribution: The TREND-linker (excluding ribose and phosphate) is in close proximity to D91, K118, and E196 of DrrA, yet no specific interactions are being observed (Supplementary Figure 7). We therefore conclude that the linker is not contributing to the interface profoundly. Also, new experiments reveal that both AMPylated and non-AMPylated Rab1b can stimulate DrrA's AMPylation activity with similar rates (Supplementary Figure 15, see also below). Hence, it is unlikely that the mild contacts formed by TREND-linker in the DrrA:Rab8a complex structure are of any biological meaning. We have added the following argument to our manuscript.

The covalently bound TREND is well defined in the electron density in the complex crystal structure (Figure 2b). However, due to structural differences of the chemical linker, bridging the TREND ribose and C197 in DrrA, compared to ATP, only few interactions with the central scaffold of the linker are formed, which is reflected by the higher B-factors in respect to the surrounding amino acid residues (Supplementary Figure 7).

Supplementary Figure 7. TREND-linker binding site. a) Surface representation of the DrrA16-352 - L197C-Trend-Rab8 complex, with the TREND-linker and surrounding residues shown as stick models. b) Ligplot diagram (Laskowski and Swindells, 2011) of the amino acid residues interacting with the TREND linker. Hydrogen bonds are drawn as black lines, residues in Van der Waals-distance are highlighted as red half circles and the covalent bonds are shown in cyan. c) Stereo view of the TREND-linker (pink) and its Fo-Fc simulated annealing omit-electron density map (blue mesh, contoured at 2σ) and surrounding amino acids. (DrrA: grey; Rab8: blue, TREND-linker: pink)

Regarding the membrane facing side of Rab1: Rab1 can bind to the *Legionella* containing vacuole (LCV) via its lipidated C-terminus. At the same time, the C-terminal phosphatidylinositol-4-phosphate (PI4P) binding domain of DrrA interacts with PI4P lipids in the LCV. By superimposing the current and previous complex structures, we have created a full-length DrrA model (Figure 5e). Also, we have updated our final schematic model of DrrA-mediated AMPylation in order to demonstrate the binding events as they may occur in a membrane context (Figure 5f).

Figure 5e. Composite model of DrrA produced from the following crystal structures PDB: 6YX5, 3LOI, 3N6O. The different binding sites of Rab-molecules are indicated at the GEF-domain and the NC-RBS. The catalytic site of the AMPylation domain (composed by D110, D112, D150) is indicated by magenta spheres. GEF: Guanine nucleotide exchange factor; P4M: phosphatidylinositol-4-phosphate (PI4P) binding of SidM/DrrA.

Figure 5f. Model of DrrA action at the membrane. The P4M localizes DrrA to the membrane by binding to PI4P. The GEF-domain activates Rab1 by GDP-to-GTP exchange, thereby displacing it from GDI and creating membrane-bound Rab1:GTP (yellow ribbons indicate geranylgeranyl lipids). Binding of Rab1:GTP to the NC-RBS of DrrA putatively results in structural rearrangement of the catalytic site, which stimulates the subsequent AMPylation of another Rab1:GTP-molecule.

In response to the referee comments, we have amended our discussion as follows:

Also, the structure of the DrrA_{ATase} permits us to produce a composite structural model by superimposing previous structures (Muller et al., 2010; Zhu et al., 2010), in which 3 different Rab1 binding sites can be visualized (Figure 5e). The NC-RBS profoundly contributes to the regulation of the AMPylation activity: GTP-loaded Rab1b binds to an allosteric site located opposite the catalytic centre of the AMPylation domain, and stimulates AMP transfer to the GTPase. Thus, two molecules of Rab1b:GTP simultaneously bind to DrrA; one to the allosteric NC-RBS, the other to the catalytic AMPylation site. Thus, DrrA binds to the LCV by virtue of the PI4P binding P4M and catalyses the activation of Rab1 via GDP-to-GTP-exchange using its GEF-domain. This displaces Rab1 from GDI and leads to membrane binding through the C-terminally attached geranylgeranyl-lipids. Subsequently, Rab1:GTP stimulates the AMPylation activity of DrrA by binding to the NC-RBS of the ATase-domain, leading to AMPylation of other recruited Rab1-molecules (see model in Figure 5f).

Comment 3

An important question is whether the backside Rab1-binding site can accommodate/be regulated by unmodified Rab1-GTP (which is the product of the GEF reaction), AMPylated Rab1-GTP (which is the product of the AMPylation reaction), or both. An experiment clarifying this issue would provide clues as to the sequence of events in the context of the entire protein and would greatly increase the significance of the study. In addition, the authors should show the docking of unmodified Rab1-GTP and of AMPylated Rab1 within the crosslinked Rab8-DrrA structure to identify possible structural conflicts.

We thank the reviewer for the critical suggestion. In order to address whether unmodified Rab1:GTP or AMPylated Rab1:GTP binds to the NC-RBS of DrrA, we designed an assay in which crosslinking obtained by genetic code expansion is used as a tool to monitor and distinguish both unmodified and AMPylated Rab1 binding. In this assay, we have used a catalytically inactive crosslinkable DrrA construct (DrrA₁₆₋₃₅₂_D110A_D112A_D82C) and either unmodified or pre-AMPylated GTP hydrolysis-deficient crosslinkable UAA Rab1 construct (Rab1-Q67A_R69BrC6K). This approach allowed us to assess the binding of AMPylated and unmodified Rab1 to the NC-RBS without the interference of catalytic activity of DrrA. To this end, we analyzed the crosslinked complex formation of DrrA:Rab1 or DrrA:Rab1-AMP via western-blot at different time points. Supplementary Figure 15 shows that there is no significant difference between unmodified Rab1 and AMPylated Rab1 in terms of efficiency of the formation of covalent complex with DrrA at the NC-RBS. Given the fact that we observe no cytotoxicity from DrrA₈₋₅₃₃_R70A/Q71A/K74A since Rab1 binding to NC-RBS of this mutant is hindered by mutations (Figure 4g), our results suggest that unmodified Rab1 should initially turn the catalytic activity of DrrA_{ATase} on. After the emergence of activated DrrA molecules on the LCV by unmodified Rab1:GTP, both unmodified and AMPylated Rab1:GTP may involve in activation of other DrrA molecules on the LCV.

To account for this observation, we have added the following paragraph to the result section of our manuscript:

In addition, we wondered whether the presence of the AMP-group in Rab1b at Y77_{Rab1b} would exclude its binding to the NC-RBS. We therefore utilized our established approach and monitored time-resolved crosslink formation (via D82C_{DrrA}) of either AMPylated or non-AMPylated Rab1b-R69BrC6K-His6 with the DrrA₁₆₋₃₅₂-D82C-D110A-D112A variant by SDS-PAGE and western-blotting. The AMPylation-deficient D110A/D112A DrrA-mutant was used to exclude modification of non-AMPylated Rab1b during the experiment. Indeed, both AMPylated and non-AMPylated Rab1 form a crosslinked complex *in vitro* with minor differences in the rate of reaction (Supplementary Figure 15). Hence, the presence of AMP in Rab1b does not exclude binding to the NC-RBS.

In summary, biochemical experiments, cytotoxicity data and crosslinking approaches confirm that the AMPylation activity of DrrA is allosterically stimulated by Rab1b-binding to the NC-RBS.

We now superimposed AMPylated Rab1 with the current DrrA:Rab8 complex structure to identify possible structural clashes between the nucleobase of AMP and DrrA. However, we believe that the superimposition does not represent a likely scenario. The reason is that the position of AMPylated Tyr77 in Rab1b is not known and cannot be deduced from the crystal structure, since the AMPylated side chain is directly involved in crystal contact formation. Therefore, there is a significant chance that the observed orientation of Tyr77-AMP_{Rab1b} is artificial and not reflecting the situation in solution or bound to DrrA. This notion is supported by our findings outlined in Supplementary Figure 15: The seeming clashes between DrrA and the AMP-group in Rab1b at Y77 would be incompatible with Rab1b-binding to the NC-RBS of DrrA. Yet, AMPylated Rab1b can still form a complex with DrrA (see above). Therefore, this superimposition does not contribute to understanding the influence of the AMP-group for DrrA-binding.

We also interpreted the suggestion of the reviewer alternatively. As for the reviewer's suggestion for the structural conflicts, we analyzed whether amino acid side chains of Rab1b may sterically interfere with binding to DrrA when being superimposed on the structure of Rab8a in the DrrA:Rab8a-complex. Most amino acids are identical in these Rab proteins; however, differences are observed for residues located in the interface: 26, 29, 32 and 35 (Rab1b: L26, A29, T32, E35 vs. Rab8a: F26, S29, A32, S35). None of these amino acids contribute to the interactions in the complex interface (see Figure 2c and Supplementary Figure 9). Also, these Rab1 residues do not produce steric clashes with DrrA. These observations are in line with *in vitro* AMPylation data: DrrA AMPylates Rab1b and Rab8a with comparable rates (Supplementary Figure 5a). We have added the following text and figures to our manuscript in order to address the reviewer comment:

Supplementary Figure 5. a) Catalysis kinetics of DrrA mutants toward Rab8a: GppNHp. The k_{cat}/K_M value of wt DrrA is $9.0 \times 10^5 \text{ M}^{-1} \text{ s}^{-1}$ ($\pm 1.2 \times 10^5 \text{ M}^{-1} \text{ s}^{-1}$). Data are means \pm standard error of the mean (SEM) from three independent experiments. b) Catalysis kinetics of DrrA toward Rab1b:GppNHp and Rab8a:GppNHp. The k_{cat}/K_M value of wt DrrA₁₆₋₃₅₂ toward Rab8a:GppNHp is $9.0 \times 10^5 \text{ M}^{-1} \text{ s}^{-1}$ ($\pm 1.2 \times 10^5 \text{ M}^{-1} \text{ s}^{-1}$). The k_{cat}/K_M value of wt DrrA₁₆₋₃₅₂ is $6.2 \times 10^5 \text{ M}^{-1} \text{ s}^{-1}$ ($\pm 2.0 \times 10^4 \text{ M}^{-1} \text{ s}^{-1}$). Data are means \pm standard error of the mean (SEM) from three independent experiments.

Since the complex was obtained with Rab8a as a Rab1b-surrogate, we tested whether there are notable differences of the mode of DrrA-binding. Structural superimposition of Rab1b with Rab8a from the DrrA:Rab8a complex revealed four amino acids that are located in the interface and differ between the Rab-proteins: Rab1b (Muller et al., 2010): L26, A29, T32, E35 vs. Rab8a: F26, S29, A32, S35. However, none of these amino acids are involved in interactions in the complex interface and do not create steric clashes that could impair complex formation between DrrA₁₆₋₃₅₂ and Rab1b (Figure 2c, Supplementary Figure 9). Consistent with the lack of steric clashes and the high sequence homology between Rab1b and Rab8a, the AMPylation rates of these GppNHp-loaded Rabs by DrrA are comparable (Supplementary Figure 5a). Thus, the mode of binding of DrrA₁₆₋₃₅₂ to Rab1b:GppNHp and Rab8a:GppNHp is identical.

Supplementary Figure 9. Amino acid differences between Rab1b and Rab8a in the Rab-DrrA-interface. a) Superimposition of Rab8a with Rab1b in the Rab8a:DrrA-complex. Rab1b amino acid side chains that differ from Rab8a do not cause structural clashes, explaining why both Rabs are AMPylated by DrrA₁₆₋₃₅₂ *in vitro*.

Comment 4

Likewise, another important question is whether the proposed regulation of the ATase domain by Rab-GTP also takes place in full length DrrA (which carries the GEF and phospholipid-binding domains in C-terminus of the ATase domain). Kinetics experiments clarifying this issue would increase the significance of the proposed feedforward effect.

We thank the reviewer for the critical suggestion. We have now performed the kinetic study with the full-length DrrA₁₆₋₆₄₇ toward to GppNHp loaded Rab1. Indeed, analysis of the data via the sigmoidal Hill-type function indicated that allosteric binding also takes place in the full-length DrrA. New results are inserted into Figure 4e and the text has been amended as follows:

DrrA₁₆₋₆₄₇ containing all three domains shows an identical kinetic profile ($n = 1.6 \pm 0.20$) (Figure 4e).

Figure 4e. Full-length DrrA₁₆₋₆₄₇ mediated AMPylation. The red curve represents the Hill fit with a cooperativity parameter of $n = 1.6 \pm 0.20$; the black curve is the Michaelis-Menten fit. Data are means \pm SEM from *three* independent experiments.

Comment 5

The biochemical AMPylation experiments should be described more accurately. Notably, controls should be shown that the tryptophan fluorescence signal distinguishes between Rab1 AMPylation and binding of Rab1 to the allosteric site. Also, how are k_{cat}/K_m determined from single rate measurements? In addition, representative kinetics profiles should be shown for Figure 3b (mutants) and Figure 4d (allosteric activation).

As requested by the reviewer, we performed the suggested control experiments. In the absence of ATP, DrrA does not cause any change in the Rab1 tryptophan fluorescence signal. Thus, AMPylation of Rab1, but not the bindings of Rab1 to the allosteric site of DrrA, results in a change in tryptophan fluorescence signal. Control experiments are inserted as Supplementary Figure 10:

Supplementary Figure 10. Representative fluorescence traces and controls for the kinetics of Rab-AMPylation. a) Negative control for enzyme kinetics of DrrA₁₆₋₃₅₂ mediated Rab1b:GppNHp AMPylation based on time-resolved tryptophan fluorescence. b) Positive control for enzyme kinetics of DrrA₁₆₋₃₅₂ mediated Rab1b:GppNHp AMPylation based on time-resolved tryptophan fluorescence. c) Negative control for enzyme kinetics of DrrA₁₆₋₃₅₂ mediated Rab8a:GppNHp AMPylation based on time-resolved tryptophan fluorescence. d) Positive control for enzyme kinetics of DrrA₁₆₋₃₅₂ mediated Rab8a:GppNHp AMPylation based on time-resolved tryptophan fluorescence. The tryptophan fluorescence intensity of Rab proteins are normalized to 1.0. All panels: Rab1b₃₋₁₇₄, Rab8a₆₋₁₇₆, and DrrA₁₆₋₃₅₂ are referred to as Rab1b, Rab8a and DrrA, respectively.

For the determination of catalytic efficiencies (k_{cat}/K_M) of AMPylation reactions measured by fluorescence spectrometry, reaction curves were fitted to a single exponential curve using the following equation:

$$F(t) = F_0 + F_A \cdot e^{-k_{obs} \cdot t}$$

Where $F(t)$ is the fluorescence intensity, F_0 is the minimum fluorescence intensity, F_A is the total fluorescence amplitude (i.e., $F_{max} - F_0$, with F_{max} as the maximum fluorescence intensity), and k_{obs} is the observed rate constant.

The observed rate constant (k_{obs}) was divided by the applied DrrA concentration (100 nM), yielding k_{cat}/K_M .

As requested by the reviewer, we also inserted the representative kinetics profiles into Supplementary Figure 17.

Also, we have added the following text to our *Materials and Methods* section:

In brief, for the determination of catalytic efficiencies (k_{cat}/K_M) of AMPylation reactions measured by fluorescence spectrometry, reaction curves were fitted to a single exponential curve according to using the following equation:

$$F(t) = F_0 + F_A \cdot e^{-k_{obs} \cdot t}$$

Where $F(t)$ is the fluorescence intensity, F_0 is the minimum fluorescence intensity, F_A is the total fluorescence amplitude (i.e., $F_{max} - F_0$, with F_{max} as the maximum fluorescence intensity), and k_{obs} is the observed rate constant. The observed rate constant (k_{obs}) was divided by the applied DrrA concentration (100 nM), yielding k_{cat}/K_M .

Comment 6

Cytotoxic effects have been measured in cells with the 70-71-74 DrrA triple mutant. To fully correlate these effects with the *in vitro* AMPylation experiments, the authors should show the toxicity of the

same single mutants (57, 63, 64, 70, 71, 74, 194, and 195) in cells, which are mentioned but not shown.

As requested by the reviewer, we now repeated cytotoxicity study of the single mutants (Figure 4f). Our *in vitro* kinetic study indicated that R70A, Q71A, and K74A significantly impair the AMPylation activity of DrrA. Therefore, we selected these mutants for cytotoxicity studies in H1299 cells. However, the selected point mutations cause almost identical cytotoxicity as wt DrrA (Figure 4f). Although kinetics indicated that these single mutants dramatically decrease the enzyme activity *in vitro*, we believe that the situation *in cellulo* may be very different due to the extended time of cellular experiments. To make an approximate comparison, we further monitored Rab1 AMPylation by the single DrrA mutants R70A_{DrrA}, Q71A_{DrrA} or K74A_{DrrA} via intact high resolution mass spectrometry at 72 hours incubation time. *In vitro* experiments with extended incubation time of single mutants resulted a similar outcome as expected (Supplementary Figure 13).

Structural analysis indicated that R70, Q71, and K74 need to interact with each other to constitute a triangle- like network, which can further stabilize the interactions between them and D44 from Rab1 or Rab8a. Therefore, we hypothesized that the triple mutant DrrA₈₋₅₃₃R70A/Q71A/K74A may result significant difference in cytotoxicity compared with wt DrrA₈₋₅₃₃. Indeed, as seen by Figure 4g, triple mutant causes no cytotoxicity as expected. Added the following text and figure to the manuscript to describe this observation:

DrrA₈₋₅₃₃ but not the AMPylation-deficient D110A/D112A_{DrrA} mutant caused pronounced cytotoxicity¹⁴. However, H1299 cells transfected with DrrA₈₋₅₃₃ or these selected single alanine mutant R70A_{DrrA}, Q71A_{DrrA} or K74A_{DrrA} showed similar cytotoxicity (Figure 4f). Although kinetics indicated that single alanine mutants (R70A_{DrrA}, Q71A_{DrrA} or K74A_{DrrA}) dramatically decrease the enzyme activity *in vitro*, results from experiments in cellulo differ from *in vitro* studies probably due to the longer procedure of the *in vivo* experiments. To make an approximate comparison, we investigated Rab1b-AMPylation by the DrrA single mutants R70A_{DrrA}, Q71A_{DrrA} or K74A_{DrrA} via intact high resolution mass spectrometry at 72 hours incubation time. Extended incubation times of the single mutants resulted in nearly quantitative AMPylation of Rab1b(Supplementary Figure 13). Structural analysis of the DrrA:Rab8a-complex indicated that these three residues can interact with each other to constitute a triangular network, which may stabilize the interaction with D44 from Rab1b/Rab8a. We therefore reasoned that the simultaneous combination of these mutations is required to result in a reduction in cytotoxicity compared to wt DrrA₈₋₅₃₃.

Figure 4f. Cytotoxicity analysis of DrrA single alanine mutants in H1299 cells. Cell viability values (determined by MTS assay) of DrrA-expressing cells (eGFP-positive) were determined in relation to the eGFP vector control. WT: DrrA₈₋₅₃₃; R70A: DrrA₈₋₅₃₃_R70A; Q71A: DrrA₈₋₅₃₃_Q71A; K74A: DrrA₈₋₅₃₃_K74A.

Supplementary Figure 13. *In vitro* AMPylation of 100 μ M Rab1b₃₋₁₇₄:GppNHp by DrrA₁₆₋₃₅₂ (5 μ M) and DrrA NC-RBS-mutants (5 μ M). AMPylation was carried out for 72 h. The mass of AMPylated and non-AMPylated Rab1b₃₋₁₇₄ was determined by intact mass spectrometry. The triple-mutant R70A/Q71A/K74A has minor AMPylation activity, whereas the single mutants are still capable of producing AMPylated Rab1b (Mw(Rab1b₃₋₁₇₄)= 19,729 Da, Mw(AMP-Rab1b₃₋₁₇₄)=20,059 Da. a) Rab1b AMPylated by wt DrrA₁₆₋₃₅₂. b) Rab1b AMPylated by DrrA₁₆₋₃₅₂R70A/Q71A/K74A. c) Rab1b AMPylated by DrrA₁₆₋₃₅₂R70A. d) Rab1b AMPylated by DrrA₁₆₋₃₅₂Q71A. e) Rab1b AMPylated by DrrA₁₆₋₃₅₂K74A.

Comment 7

Could the authors comment on the physiological relevance of the range of Rab concentrations used in Figure 4d (allosteric activation, up to 250 μM)?

The referee probably expresses his/her doubt that a Rab-concentration of 250 μM is physiologically relevant. We share this view. We believe that we cannot reliably comment on the concentration these particular proteins in the cell. The matter is even more complicated since these DrrA- and Rab-dependent processes are happening on the membrane (i.e. the Legionella containing vacuole) and not in the cytosol. Therefore, the local concentration of DrrA and Rab1 are entirely unknown.

Even though the interaction of Rab1 with the non-conventional, allosteric Rab binding site on DrrA may appear to be of moderate affinity only, interactions with K_D values in the range of 10-200 μM are in fact common for proteins and enzymes, even though their cellular concentrations are likely much lower than this value. Likely, low affinities ensure that an interaction is dependent on and responsive to a wide range of dynamic concentrations changes of one or both proteins, thus permitting to finetune the biological output of that interaction.

However, these reflections are merely speculative, as are the comments on the potential concentrations for Rab proteins and DrrA in the cell, in the cytosol or at the membrane. Even this is an important point raised by this referee, we would like to refrain from taking this speculation to the manuscript for the reasons outlined before.

Comment 8

The proposed regulatory mechanism should take place entirely on the membrane, with the newly discovered interaction of Rab1-GTP with the ATase domain adding an anchoring point. Although assessing this regulatory mechanism in a membrane context would admittedly require too much work, could the authors at least comment on how they envision the structural organization of Rab1 and the three Rab1-binding sites of DrrA on the surface of the LCV, and the sequence of events? A schematic view would be helpful.

We appreciate the reviewer's suggestion; therefore, we have now created a schematic representation of the Rab1 AMPylation taken place on the LCV during infection to simplify the understanding the sequence of events (Figure 5f). As illustrated in Figure 5f, Rab1: GDP will be recruited to LCV by the GEF activity of DrrA. Subsequently, this Rab1:GTP will bind to the allosteric site (the same site as Rab1 binding site in GEF) of the AMPylation domain. Then DrrA is activated and AMPylates Rab1. The following alterations have been made to text and figures:

The NC-RBS profoundly contributes to the regulation of the AMPylation activity: GTP-loaded Rab1b binds to an allosteric site located opposite the catalytic centre of the AMPylation domain, and stimulates AMP transfer to the GTPase. Thus, two molecules of Rab1b:GTP simultaneously bind to DrrA; one to the allosteric NC-RBS, the other to the catalytic AMPylation site. Thus, DrrA binds to the LCV by virtue of the PI4P binding P4M and catalyses the activation of Rab1 via GDP-to-GTP-exchange using its GEF-domain. This displaces Rab1 from GDI and leads to membrane binding through the C-terminally attached geranylgeranyl-lipids. Subsequently, Rab1:GTP stimulates the AMPylation activity of DrrA by binding to the NC-RBS of the ATase-domain, leading to AMPylation of other recruited Rab1-molecules (see model in Figure 5f).

Figure 5f. Model of DrrA action at the membrane. The P4M localizes DrrA to the membrane by binding to PI4P. The GEF-domain activates Rab1 by GDP-to-GTP exchange, thereby displacing it from GDI and creating membrane-bound Rab1:GTP (yellow ribbons indicate geranylgeranyl lipids). Binding of Rab1:GTP to the NC-RBS of DrrA putatively results in structural rearrangement of the catalytic site, which stimulates the subsequent AMPylation of another Rab1:GTP-molecule.

Comment 9

The proposed structural mechanism for allosteric activation of DrrA is highly speculative and alternative explanations, for example a hinge motion induced by Rab1 to reposition active site residues contributed by the two subdomains, cannot be excluded. This section should be toned down.

We understand the reviewer's concern. As discussed in the comment 1, structural integrity of the whole ATase domain and allosteric Rab1 binding are important for the catalytic activity. Since only few interactions are observed between the C-terminal part and the catalytic site residues in the N-terminal part of the ATase domain, the contribution of the allosteric Rab1 binding to catalytic activity of DrrA is significant. Furthermore, the kinetic study by DrrA₁₆₋₆₄₇ and DrrA₁₆₋₃₅₂ constructs directly proves an allosteric mechanism, supporting the role of Rab1 binding to NC-RBS. As brought out by the reviewer, we have adjusted our discussion accordingly to highlight the role of Rab1 binding by not excluding the contribution of C-terminal part in the structural integrity of DrrA ATase domain. We have adjusted our text as follows:

In summary, biochemical experiments, cytotoxicity data and crosslinking approaches confirm that the AMPylation activity of DrrA is allosterically stimulated by Rab1b-binding to the NC-RBS. However, to what extent the binding of Rab1b to the NC-RBS leads to a structural reorganization of the catalytic site cannot be deduced entirely from the available structure comparisons: In the absence of the C-terminal part of the ATase-domain (referred to as part-C), essential intramolecular interactions between part-C and part-N are lacking and thus the catalytic residues may be more flexible than in a –yet unavailable– ATase reference structure. Thus it is possible that the impact of the here observed structural changes introduced by Rab1b on the ATase-domain of DrrA are less pronounced in context of the full-length DrrA protein.

Comment 10

I am not convinced by the proposed “safety” mechanism, as the switch 2 of cytosolic Rab GTPases is masked by their interactions with RabGDI and is thus unlikely to be able to activate DrrA. Could alternative functions be envisioned?

Indeed, the safety mechanism is highly speculative. The proposition reflects our attempt to make sense of the allosteric activation by DrrA in terms of the infection process. But the referee is correct that GDP-bound Rabs are bound to GDI in the cytosol and thus switch II is inaccessible to DrrA for AMPylation. However, even though switch II is blocked, it must be accessible to binding partners for a limited amount of time. Otherwise, activation of Rab-proteins could never happen since also GEFs require binding to switch II. Consequently, the binding of Rabs and GDI must be (weakly) dynamic to permit binding of GEFs and thus Rab activation. Therefore, accidentally cytosolic DrrA could – if unregulated – AMPylate Rab1 after spontaneous dissociation of GDI and therefore deplete the functional cellular Rab1-pool. The proposed safety mechanism would solve this problem since the ATase domain would only obtain full activity in the presence of Rab1b:GTP that has been produced by the action of the GEF-domain of DrrA. But, certainly, this view can only be hypothetical. We have therefore added clarifications and a statement of caution to the manuscript in the discussion section:

However, in the GDP-form, Rab-proteins are complexed with GDI in the cytosol, thereby making switch II inaccessible for AMPylation by DrrA. But, the Rab1:GDI complex is weakly dynamic because cellular GEFs are known to contact switch II, leading to GDI-displacement by loading the Rab with GTP. Thus, it is conceivable that DrrA could also AMPylate GDP-bound Rab whenever it has been spontaneously dissociated from GDI, explaining the need for an additional safety mechanism, i.e. allosteric activity control of DrrA-AMPylation activity by Rab1b:GTP-binding. Nevertheless, this hypothesis remains speculative at this point.

Minor point: the concentration units in the kinetics methods section contain some errors.

We thank the reviewer for bringing this to our attention. We corrected the concentration units from mM to nM.

Reviewer #2

In this study, Itzen and colleagues continue their biophysical characterization of DrrA (a.k.a. SidM) in its interactions with its substrate Rab1b. The extensive use of biochemical and biophysical developed by the authors, including two different approaches that explore proximity-triggered and site-specific chemical crosslinking for structural analysis of DrrA domains in complex with Rabs has generated some unique insights into the mechanism of catalysis of its AMPylation activity, including the proposal that Rab1b allosterically activates DrrA. Most of the experiments were very well performed, but the evidence for some major conclusions needs further to be substantiated.

We thank the reviewer for the positive reception of our study.

Specific comments

Comment 1

It is understood that the use of Rab8a for the complex structure is to circumvent the technical challenge in obtaining useful crystals with the bona fide substrate Rab1b. The authors showed that DrrA can AMPylate Rab8a *in vitro*. Because Rab81 is not the natural substrate it is necessary to compare the catalysis kinetics of DrrA toward Rab1b and Rab8a. The activity of some mutants showing in Fig. 3 toward these two Rab proteins should also be examined and compared.

As requested by the reviewer, the kinetics of Rab1b and Rab8a AMPylation by DrrA₁₆₋₃₅₂ were compared (Supplementary Figure 5a). Rab8a:GppNHp is AMPylated by DrrA₁₆₋₃₅₂ *in vitro* with very similar kinetics compared to Rab1:GppNHp. In parallel, we performed the catalysis kinetics of DrrA toward Rab8a:GppNHp (Supplementary Figure 5b). In response to a comment 3 of reviewer #1, we have added to following text to the manuscript that also applies to this comment of reviewer #2:

Since the complex was obtained with Rab8a as a Rab1b-surrogate, we tested whether there are notable differences of the mode of DrrA-binding. Structural superimposition of Rab1b with Rab8a from the DrrA:Rab8a complex revealed four amino acids that are located in the interface and differ between the Rab-proteins: Rab1b: L26, A29, T32, E35 vs. Rab8a: F26, S29, A32, S35. However, none of these amino acids are involved in interactions in the complex interface and do not create steric clashes that could impair complex formation between DrrA₁₆₋₃₅₂ and Rab1b (Figure 2c, Supplementary Figure 9). Consistent with the lack of steric clashes and the high sequence homology between Rab1b and Rab8a, the AMPylation rates of these GppNHp-loaded Rabs by DrrA are comparable (Supplementary Figure 5a). Thus, the mode of binding of DrrA₁₆₋₃₅₂ to Rab1b:GppNHp and Rab8a:GppNHp is identical.

Supplementary Figure 5. a) Catalysis kinetics of DrrA mutants toward Rab8a: GppNHp. The k_{cat}/K_M value of wt DrrA is $9.0 \times 10^5 \text{ M}^{-1} \text{ s}^{-1}$ ($\pm 1.2 \times 10^5 \text{ M}^{-1} \text{ s}^{-1}$). Data are means \pm standard error of the mean (SEM) from three independent experiments. b) Catalysis kinetics of DrrA toward Rab1b:GppNHp and Rab8a:GppNHp. The k_{cat}/K_M value of wt DrrA₁₆₋₃₅₂ toward Rab8a:GppNHp is $9.0 \times 10^5 \text{ M}^{-1} \text{ s}^{-1}$ ($\pm 1.2 \times 10^5 \text{ M}^{-1} \text{ s}^{-1}$). The k_{cat}/K_M value of wt DrrA₁₆₋₃₅₂ is $6.2 \times 10^5 \text{ M}^{-1} \text{ s}^{-1}$ ($\pm 2.0 \times 10^4 \text{ M}^{-1} \text{ s}^{-1}$). Data are means \pm standard error of the mean (SEM) from three independent experiments.

Supplementary Figure 9. Amino acid differences between Rab1b and Rab8a in the Rab-DrrA-interface. a) Superimposition of Rab8a with Rab1b in the Rab8a:DrrA-complex. Rab1b amino acid side chains that differ from Rab8a do not cause structural clashes, explaining why both Rabs are AMPylated by DrrA₁₆₋₃₅₂ *in vitro*

Regarding the DrrA mutants: DrrA mutants showed similar AMPylation activity profile toward to GppNHp-bound Rab8a compared to Rab1b (Supplementary Figure 5a). The AMPylation activity of the E64A_{DrrA} mutant toward Rab1 and Rab8a is the only noticeable difference. Examining the crystal structure, E64A_{DrrA} is interacting with N34 in Rab8a, but in Rab1b this amino acid is T34 (Figure 2c). This may produce the differences in enzymatic activity. We have adjusted the manuscript as mentioned above and are now mentioning the AMPylation of Rab8a by DrrA-mutants, too.

Comment 2

The physiological relevance of the proposed mechanism of catalysis should be examined. The AMPylator activity of DrrA has been shown to be essential for the recruitment of Rab1 to the LCV (PMID: 24520063). It is necessary to introduce some of the mutants in Fig. 3 and Fig. 5b into the *Legionella drrA* mutant and examine their ability in Rab recruitment.

Uncovering the physiological relevance of the proposed mechanism of catalysis will surely be interesting for the future studies. However, it is out of the scope in the current manuscript. Here, we reported that the surprising allosteric binding of active Rab1 regulates the AMPylation activity of DrrA. Therefore, it will be interesting to further investigate these mutants' ability of the recruitment of Rab1 as the part of the downstream signaling in the context of infection in a future study as a stand-alone publication.

Comment 3

The GEF domain of DrrA also contributes to its cytotoxicity to mammalian cells. Results in Fig 5E may have overestimated the effects of the AMPylation activity.

We understand the reviewer's concern. However, we have not observed cytotoxicity of DrrA_{GEF} domain in our MTS assay which was repeated multiple times (biologically independent experiments) with similar results. Overall, our results indicate that AMPylation competent construct causes severe cytotoxicity while AMPylation deficient constructs do not cause cytotoxicity on a severe level.

Comment 4

The proposed allosteric activation of DrrA by Rab1 is solely supported by conformational changes observed in steady state structures. It is necessary to examine the dynamics of the Rab-induced conformational changes.

We realize that we have not made this point clear enough in our manuscript: Our proof of having an allosteric activation mechanism of DrrA by Rab1 is not based on the structural changes suggested by the individual crystal structures of free and complexed DrrA. Rather, the structural differences in these structures tempted us to consider an allosteric activation mechanism as a hypothesis that required direct experimental proof. We therefore turned to classical biochemistry in which we used our established AMPylation assay that exploits a tryptophan fluorescence change upon covalent AMP-attachment to Rab1 to probe whether Rab1 could allosterically activate DrrA. We therefore have performed classical Michaelis-Menten-kinetics: The rate (v) of Rab1-AMPylation was determined in dependence of substrate concentration (i.e. $c(\text{Rab1:GppNHp})$) at constant DrrA-concentration (E_0) (please see Figure 4d for the result of this analysis). A plot of v (or v/E_0) over $c(\text{Rab1:GppNHp})$ would result in a hyperbolic curve according to the Michaelis-Menten-Equation (please compare the hypothetical black line in Figure 4d indicating Michaelis-Menten-behavior of the enzyme). However, it is clearly seen that the data are best fitted to a sigmoidal function rather than a hyperbolic one. This is unambiguous proof that the substrate itself (i.e. Rab1b:GppNHp) has a stimulating effect on the enzyme activity (i.e. the DrrA AMPylation activity). Therefore, this experiment demonstrates an allosteric activation of DrrA by Rab1. The crystal structures, however, merely suggest a potential mechanism how the allostery is structurally realized, yet there is no proof that the mechanism is operating truly in this manner. This, indeed, would require further work.

A detailed analysis of the dynamic changes occurring in DrrA after Rab1-binding to the allosteric site would require better suited methods such as nuclear magnetic resonance (NMR). However, NMR-experiments would require successful isotopic labelling and the assignment of the peaks to the 340 amino acids in the AMPylation domain of DrrA. In our personal experience gained in a recently

published work, establishing the labelling procedure and assigning the NMR-peaks would add a significant amount of time to the project. Albeit undoubtedly interesting, these data would not prove that allosteric activation is occurring but would instead show that structural changes may occur during Rab1 binding. However, since Rab1 can bind to the conventional (catalytic) site and the non-conventional site, distinguishing between these possibilities will be challenging if not impossible.

We hope that the referee can be content with our biochemical proof of the allosteric mechanism.

Comment 5

The role of DrrA in Legionella virulence was overstated throughout the text. Deletion of drrA did NOT cause discernable defects in bacterial virulence in all infection model tested (Refs 11 and 12). In fact, the drrA gene is absent in *L. pneumophila* strains such as Paris. Some of the references are not accurate. For example, lines 446-447, neither ref 29 nor 30 shows results to demonstrate the importance of SidJ in *L. pneumophila* infection. The study by Gan et al (PMID: 31330531) and an earlier study by Liu and Luo showed that (PMID: 17101649).

We thank the referee for mentioning this oversight. We have added the following sentence addressing the non-essentiality of DrrA for Legionella infection to the manuscript:

However, the deletion of DrrA has little impact on successful infection by *Legionella*, probably due to effector redundancy (Machner and Isberg, 2006).

Additionally, Refs 29 and 30 were changed as requested (Now Ref# 31 and 32):

31. Liu YC, Luo ZQ. The *Legionella pneumophila* effector SidJ is required for efficient recruitment of endoplasmic reticulum proteins to the bacterial phagosome. *Infection and immunity* 75, 592-603 (2007).

32. Gan N, et al. Regulation of phosphoribosyl ubiquitination by a calmodulin-dependent glutamylase. *Nature* 572, 387-391 (2019).

Comment 6

Line 41 "Legionnaires".

We thank the reviewer for spotting a typo and we gladly changed the text accordingly.

Line 41 Legionnaires was changed to Legionnaires'.

Comment 7

Somewhere in page 4 or 5, the author should discuss the role of the AMPylation activity in Rab1 recruitment to the LCV as shown by Hardiman and Roy (PMID: 24520063).

According to the reviewer's suggestion, we changed the text and cite the references accordingly. The text now reads as follows:

Notably, it was reported that GFF- defective DrrA mutants are competent in the recruitment of Rab1 from Rab1: GDI complex to the LCV (Hardiman and Roy, 2014).

Comment 8

Line 446, "SidE" should be "members of the SidE family" and the authors should reference the study that originally demonstrated the ubiquitin ligase activity of these proteins.

The error has been corrected as suggested.

Reviewer #3

The manuscript “Rab1-AMPylation by Legionella DrrA is allosterically activated by Rab1” by Du, et al., describes the interaction of Legionella pneumophila DrrA with the host (human) Rab1. Traditional structural identification of the complex failed. However, through amber suppression mutagenesis with crosslinking amino acids and analysis of the resulting complex through MS/MS, x-ray crystallography, and mutagenesis studies, the authors conclude that a non-conventional Rab binding site is used for DrrA binding, resulting in allosteric control of AMPylation as a way to suppress host innate immunity. The crystal structure is solved to sufficient resolution and R_{free} levels to correctly identify the interaction interface, although resolution of the linker (Figure 2B) appears to be poor. Mutagenesis studies further corroborate the findings, although the proposed mechanism remains speculative. Two additions are suggested:

We appreciate the reviewer’s critical evaluation of our manuscript.

Comment 1

Supplementary Figure 2 is insufficient. It is large and has low resolution, and the data are not easy to interpret by a non-expert. More careful discussion and analysis of the location of crosslinking is critical to this study and should be provided.

We agree that the initial identification of the DrrA_{ATase}:Rab1b crosslink is a key step in the manuscript and that the methods used should be comprehensible. We revisited Supplementary Figure 2 and decided to remove the focus from a single peptide and rather explain the full scope of the analysis. To this end, first, we added a Supplementary Note on the mass-spectrometry based crosslink detection. The note extends the information in the Methods section in explaining the different challenges we faced in respect to this extraordinary analysis. It also helps non-proteomics experts to follow our decision rationales. Second, we added Supplementary Table 1 to show all three identified crosslink spectras between DrrA and Rab1. Third, we included Supplementary Figure 2, which sets these three measurements into the context of the full dataset and thus allowed to estimate the confidence of these identifications. Of course, we added the full analysis output to the PRIDE repository – including the data to review all of the raw mass spectra, if this is of interest to our readers (Supplementary Table 1).

Supplementary note on the analysis of mass-spectrometric crosslink data

In the mass-spectrometry based detection of crosslinks between DrrA_{ATase} and Rab1b, we faced the challenge to find a very small portion of crosslinked peptides in an overwhelming background of unmodified peptides. Several improvement steps in the biochemical methodology finally made it possible to identify crosslinked peptides between the two target proteins. These peptides were detected using high-precision mass spectrometry coupled to the crosslink detection software Kojak (Hoopmann et al., 2016; Hoopmann et al., 2015).

In order to account for human protein contamination, we appended a list of common contaminants to the DrrA_{ATase}/Rab1b search database and applied a target-decoy strategy based on reversed sequences to estimate the fraction of false positives in the output (see Methods section). For all of these interprotein crosslink events, we identified 4,998 putative crosslinks and ~73% of these crosslinks bore either one or two decoy peptides. In the remaining ~27% of identifications only three could be attributed to the DrrA_{ATase}:Rab1b complex (Supplementary Table 1), whereas the others were crosslinks involving contaminant peptides.

As each identification comes with an assigned score, we then visualized the score distributions for decoy and target identifications to get a sense for the quality of the DrrA_{ATase}:Rab1b peptides in this context (Supplementary Figure 2). The plot clearly indicates that these crosslinks of interest fall in a score range, where the target distribution is outperforming the decoy distribution. Additionally, the two best-scoring hits were assigned to the same crosslink site. We are aware of the fact that these spectra were acquired right after each other and thus likely stem from the same peptide, but given the notion that we did this analysis to get an initial idea, where a crosslink could happen, we took all this evidence and moved on to biochemically validate the finding.

Comment 2

B-factors of components of the crosslinked complex (not average) should be analyzed and provided for the proteins and the linker to help clarify the linker resolution

We have mapped the B-factors mapped onto the ribbon diagram of the structure (Supplementary Figure 6), plots showing the B-factors of the main chain atoms per residues. Additionally, we present a stereo figure and detailed ligplot diagram of the TReND-linker interactions (Supplementary Figure 7). Since there are, apart from the covalent bonds, only few interactions with the central scaffold of the linker due to structural differences compared to ATP, there is flexibility as reflected by the higher B-factors and quality of the simulated annealing-omit electron density. The here used ATP-analogue is only used as a tool to trap and access the transient Rab-DrrA complex. Thus, we are focusing in the interaction between DrrA and Rab8a in the here trapped complex, which is also confirmed by mass spectrometry.

We have added the following description to the result section of the manuscript:

The covalently bound TReND is well defined in the electron density in the complex crystal structure (Figure 2b). However, due to structural differences of the chemical linker, bridging the TReND ribose and C197 in DrrA, compared to ATP, only few interactions with the central scaffold of the linker are formed, which is reflected by the higher B-factors in respect to the surrounding amino acid residues (Supplementary Figure 6 and 7).

References

Hardiman, C.A., and Roy, C.R. (2014). AMPylation Is Critical for Rab1 Localization to Vacuoles Containing *Legionella pneumophila*. *MBio* 5, e01035- 01013.

Hoopmann, M.R., Mendoza, L., Deutsch, E.W., Shteynberg, D., and Moritz, R.L. (2016). An Open Data Format for Visualization and Analysis of Cross-Linked Mass Spectrometry Results. *Journal of the American Society for Mass Spectrometry* 27, 1728-1734.

Hoopmann, M.R., Zelter, A., Johnson, R.S., Riffle, M., MacCoss, M.J., Davis, T.N., and Moritz, R.L. (2015). Kojak: efficient analysis of chemically cross-linked protein complexes. *Journal of proteome research* 14, 2190-2198.

Laskowski, R.A., Jablonska, J., Pravda, L., Varekova, R.S., and Thornton, J.M. (2018). PDBsum: Structural summaries of PDB entries. *Protein Sci* 27, 129-134.

Laskowski, R.A., and Swindells, M.B. (2011). LigPlot+: multiple ligand-protein interaction diagrams for drug discovery. *J Chem Inf Model* 51, 2778-2786.

Machner, M.P., and Isberg, R.R. (2006). Targeting of host Rab GTPase function by the intravacuolar pathogen *Legionella pneumophila*. *Dev Cell* 11, 47-56.

Muller, M.P., Peters, H., Blumer, J., Blankenfeldt, W., Goody, R.S., and Itzen, A. (2010). The *Legionella* effector protein DrrA AMPylates the membrane traffic regulator Rab1b. *Science* 329, 946-949.

Zhu, Y., Hu, L., Zhou, Y., Yao, Q., Liu, L., and Shao, F. (2010). Structural mechanism of host Rab1 activation by the bifunctional *Legionella* type IV effector SidM/DrrA. *Proc Natl Acad Sci USA* 107, 4699-4704.

REVIEWERS' COMMENTS

Reviewer #1 (Remarks to the Author):

The authors responded thoroughly to the reviewer's comments. I am happy with the added experiments, data, figures and elements of discussion, which greatly improve the manuscript. I remain extremely enthusiastic about the work and I highly recommend it.

Reviewer #2 (Remarks to the Author):

The authors have adequately addressed my concerns with the exception of the experiments to test how the several DrrA mutants defective in efficient Rab1 AMPylation recruit Rab1 to the bacterial phagosome. The Δ drrA mutant strain and plasmid expressing DrrA are available in the research community, and the experiment is highly straightforward. It is disappointing that the authors chose not to perform these experiments. In my opinion, results from this experiment will significantly increase the meaningfulness of their findings and rule out any concern that their biophysical and biochemical results are purely artifact occurred in test tubes.

Response to referees

Rab1-AMPylation by Legionella DrrA is allosterically activated by Rab1

Reviewer #1

The authors responded thoroughly to the reviewer's comments. I am happy with the added experiments, data, figures and elements of discussion, which greatly improve the manuscript. I remain extremely enthusiastic about the work and I highly recommend it.

We are grateful to the reviewer for supporting our manuscript.

Reviewer #2

The authors have adequately addressed my concerns with the exception of the experiments to test how the several DrrA mutants defective in efficient Rab1 AMPylation recruit Rab1 to the bacterial phagosome. The Δ drrA mutant strain and plasmid expressing DrrA are available in the research community, and the experiment is highly straightforward. It is disappointing that the authors chose not to perform these experiments. In my opinion, results from this experiment will significantly increase the meaningfulness of their findings and rule out any concern that their biophysical and biochemical results are purely artifact occurred in test tubes.

We are now addressing the concern of the reviewer in the updated version of our manuscript. As suggested by the editor, we have toned down statement regarding the potential physiological impact of the allosteric regulation mechanism. We have explicitly stated that any conclusion on the role of DrrA-activity regulation for Legionella infections must remain speculative at this time.